# Towards Real-world Debiasing: Rethinking Evaluation, Challenge, and Solution

## Abstract

Spurious correlations in training data significantly hinder the generalization capability of machine learning models when faced with distribution shifts, leading to the proposition of numberous debiasing methods. However, it remains to be asked: *Do existing benchmarks for debiasing really represent biases in the real world?* Recent works attempt to address such concerns by sampling from real-world data (instead of synthesizing) according to some predefined biased distributions to ensure the realism of individual samples. However, the realism of the biased distribution is more critical yet challenging and underexplored due to the complexity of real-world bias distributions. To tackle the problem, we propose a fine-grained framework for analyzing biased distributions, based on which we empirically and theoretically identify key characteristics of biased distributions in the real world that are poorly represented by existing benchmarks. Towards applicable debiasing in the real world, we further introduce two novel real-world-inspired biases to bridge this gap and build a systematic evaluation framework for real-world debiasing, RDBench[1]. Furthermore, focusing on the practical setting of debiasing w/o bias label, we find real-world biases pose a novel *Sparse bias capturing* challenge to the existing paradigm. We propose a simple yet effective approach named Debias in Destruction (DiD), to address the challenge, whose effectiveness is validated with extensive experiments on 8 datasets of various biased distributions.

## 1 Introduction

With the rapid development of machine learning, machine learning systems are increasingly deployed in high-stakes applications such as autonomous driving and medical diagnosis, where incorrect decisions may cause severe consequences. As a result, the robustness to distribution shift is crucial in building trustworthy machine learning systems. One of the major reasons why machine learning models fail to generalize to shifted distributions in the real world (Chu et al., 2023) is because the existence of spurious correlation, i.e. biases, in training data (Wiles et al., 2022). Spurious correlation refers to the phenomenon that two distinct concepts are statistically correlated in the training distribution, yet uncorrelated in the test distribution for there is no causal relationship between them (Chu & Li, 2023). For example, rock wall background may be correlated with the sport climbing in the training data, but they are not causally related and climbing can be indoors or on ice as well (Lee et al., 2021; Chu et al., 2021; 2020). Furthermore, such spurious correlations within the data tend to be captured during training (Nam et al., 2020), resulting in a biased model that fails to generalize to shifted distributions. This lead to the proposition of various debiasing methods in recent years.

To benchmark the effectiveness of debiasing methods, both synthetic (Reddy et al., 2021; Nam et al., 2020; Liu et al., 2021) and semi-synthetic (Lee et al., 2021; Nam et al., 2020; Lim et al., 2023) (referred to as "real-world dataset" in previous works) datasets with severe biases has been adopted as benchmarks. While the individual samples in semi-synthetic datasets are realistic as they are sampled from the real world rather than synthetic, both existing synthetic and semi-synthetic benchmarks follow some predefined biased distribution that lacks thorough consideration of *how data is truly biased in the real world*, as shown in Section 2.1. This raise the following question:

> *Does existing assumption on biased distributions align with the real world?*

---

[1]RDBench: Code to be released. Preliminary version in supplementary material for anonimized review.

This is a challenging question to answer given the complexity of biases in the real world and existing coarse-grained bias analysis measures (Section 2.2). Consequently, we first revisit the biased distribution in existing benchmarks and real-world datasets and propose a fine-grained framework for analyzing bias in datasets. Inspired by the framework proposed by Wiles et al. (2022), which assumes the data is composed of some set of attributes, we further claim that analysis of dataset bias should be conducted on the more fine-grained feature (or value) level rather than attribute level, according to our observation on real-world biases. From the claim, we further propose our fine-grained framework that disentangles dataset bias into the magnitude of bias and the prevalence of bias, where the magnitude of bias generally measures how predictive (or biased) features are on the target task and the prevalence of bias generally measures how many samples in the data contain any biased feature. Empirical analysis on 8 real-world datasets across various modalities has shown that the magnitude and prevalence of real-world biases are both low, in contrast with high magnitude and high prevalence biases assumed by existing benchmarks. In section 3, we theoretically show that two strong assumptions are implicitly held by existing high bias prevalence benchmarks, which further validates our observation that real-world biases are low in bias prevalence.

Based on the empirical and theoretical insights on real-world biases, we introduce two novel type of biases inspired by real-world applications. Due to the complexity of real-world biases, debiasing methods should be capable of handling various types of biases and other real-world challenges, such as the multi-bias setting Li et al. (2023). Thus, towards developing debiasing methods applicable in the real world, we propose a systematic evaluation framework for real-world debiasing that encompass various types of biases and settings to facilitate the debiasing field.

Furthermore, focusing on the setting of debiasing w/o bias label(Lim et al., 2023; Zhao et al., 2023), which is more practical as bias feature is expensive to annotate and sometimes even hard to notice (Li & Xu, 2021), we show that the proposed real-world biases pose a novel "*Sparse bias capturing*" challenge to the existing debiasing paradigm. Specifically, the sparse and scattered real-world biases make it difficult for existing methods to identify the unknown bias accurately, causing severe performance degradation. To tackle the challenge, we introduce a simple yet effective approach, named Debias in Destruction (DiD), that can be easily applied to existing methods. Extensive experiments on 8 datasets show that DiD significantly boosts the capability of existing methods in handling various types of biases. To sum up, this work makes the following contributions:

- **Empirical and theoretical insights on biases in the real world.** We propose a fine-grained framework for bias analysis. Based on the framework, we empirically and theoretically identified key characteristics of real-world biases, previously overlooked.

- **Systematic evaluation framework for real-world debiasing.** Based on our insights, we further propose *two novel real-world-inspired biases*. Together with the multi-bias challenge and 8 existing benchmarks, we introduce a systematic evaluation framework for real-world debiasing to facilitate the development of real-world-applicable debiasing.

- **Uncover the "Sparse bias capturing" challenge in real-world debiasing.** We show that the sparsity of real-world biases poses a unique challenge for debiasing w/o bias label.

- **A simple-yet-effective approach to address the challenge.** We propose an effective approach, which can be easily applied to existing debiasing methods. An extensive evaluation on 8 datasets of various distributions shows the effectiveness of the proposed approach.

## 2 A Fine-grained Empirical Analysis on Biased Distributions

In this section, we first revisit the biased distributions in existing debiasing benchmarks and biases in the real world. Then, we propose a new framework for analyzing the biased distributions. Finally we propose our empirical findings on the consistent patterns in real-world biases.

### 2.1 Revisiting spurious correlation in datasets

**Bias in existing benchmarks.** In the area of spurious correlation debiasing, multiple synthetic (Reddy et al., 2021; Nam et al., 2020; Liu et al., 2021) and semi-synthetic datasets (Lee et al., 2021; Nam et al., 2020; Lim et al., 2023) have been adopted to benchmark the effectiveness of the debiasing methods. Generally, those synthetic datasets first select a target attribute as the learning objective (Liu

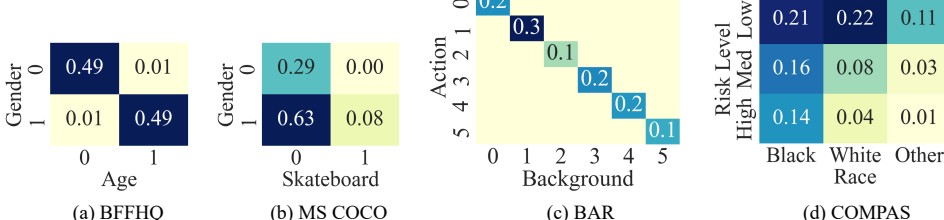

Figure 1: Visualization of the joint distribution for datasets, where the y-axis is the target attribute and the x-axis is the spurious attribute. Figure 1(a) and 1(c) visualise the distribution of existing benchmarks. Figure 1(b) and 1(d) visualize the distribution of real-world datasets. The biased distributions of existing benchmarks and real-world datasets are not alike.

et al., 2021), e.g. object, and another spurious attribute that could potentially cause the learned model to be biased, e.g. background. Then, certain sub-groups jointly defined by the target and spurious attributes, e.g. water birds with water background, are emphasized, i.e. synthesized or sampled from real-world datasets with much higher probability (usually above 95%) than the others in the biased dataset construction process, causing the corresponding spurious feature and target feature to be spuriously correlated, e.g. water background correlated with water bird (Liu et al., 2021). Specifically, one such dominating subgroup is selected for every possible value of the spurious attribute, forming a "diagonal distribution", as shown in Figure 1(a) and 1(c).

**Bias in the real world.** We further investigated biases from the real world. COCO (Lin et al., 2015) dataset is a large-scale dataset collected from the internet and widely used in various vision tasks. COCO has been found to contain gender bias in web corpora (Tang et al., 2021), one of which is the spurious correlation between males and skateboards. The joint distribution of gender and Skateboard in COCO is plotted in Figure 1(b). COMPAS (mat) dataset consists of the results of a commercial algorithm called COMPAS, used to assess a convicted criminal's likelihood of reoffending. COMPAS dataset is widely known for its bias against African Americans and is widely used in the research of machine learning fairness (Guo et al., 2023). The joint distribution of Race and Risk Level in the COMPAS dataset is plotted in Figure 1(d). Note that although COMPAS is a tabular dataset, it genuinely reflects the biased distribution in the real world. It is quite obvious that the distribution of biases in existing benchmarks and real-world datasets diverges. Additonally, CelebA is another real-world image dataset. CivilComments-WILDS (CCW) and MultiNLI are also real-world datasets in the NLP domain. *More visualizations of these additional datasets are shown in Appendix A.* In the following subsection, we will further discuss how to measure their differences.

## 2.2 PREVIOUS MEASURES OF SPURIOUS CORRELATION

We first revisit measures of spurious correlation in previous works, then point out their insufficiency.

**Background.** We assume a joint distribution of attributes $y^1, y^2, ..., y^K$ with $y^k \in A^k$ where $A^k$ is a finite set. One of these $K$ attributes is the target of learning, denoted as $y^t$, and a spurious attribute $y^s$ with $t \neq s$. The definition of spurious correlation or the measure of bias magnitude is rather vague or flawed in previous works. We summarize the measures in previous works into three categories.

**Target attribute conditioned probability.** Previous works (Wang & Russakovsky, 2023; Reddy et al., 2021) measure spurious correlation according to the probability of a biased feature $a_s$ within the correlated class $a_t$: $Corr_{tcp} = P(y^s = a_s | y^t = a_t)$. A higher value indicates a strong correlation.

**Spurious attribute conditioned probability.** Some (Tang et al., 2021; Lee et al., 2021; Yenamandra et al., 2023; Hermann et al., 2024) measure spurious correlation according to the probability of the correlated class $a_t$ within samples with biased feature $a_s$: $Corr_{scp} = P(y^t = a_t | y^s = a_s)$. A higher value of the measure indicates a strong correlation.

**Spurious attribute conditioned entropy.** Nam et al. (2020) defined an entropy-based measure of bias. They use conditional entropy to measure how skewed the conditioned distribution is: $Corr_{sce} = H(y^t | y^s)$, where $H$ is entropy. Values close to 0 indicate a strong correlation. This is an attribute-level measure, yet it is based on information theory.

We then point out the following requirements a proper measure of spurious correlation should satisfy.

***Spurious correlation is better measured at the feature level.*** As shown in Figure 1(a) and 1(c), the predictivity of every value in the spurious attribute is similar in existing benchmarks. However, this is not the case for real-world datasets, where it is clear that the predictivity of values in the spurious attribute varies greatly, as shown in Figure 1(b) and 1(d). Therefore, to deal with real-world biases, analysis of bias should be conducted on a more fine-grained value level, i.e. feature level, rather than attribute level in previous works (Nam et al., 2020). Note that though $Corr_{tcp}$ and $Corr_{scp}$ are defined at the feature level, it is assumed by previous works (Lee et al., 2021; Reddy et al., 2021; Hermann et al., 2024) that it is consistent cross features in spurious attribute during benchmark construction, i.e. viewed as an attribute level measure.

***The spurious attribute rather than the target attribute is given as a condition.*** It is well recognized that the spurious attribute should be easier than the target attribute for the model to learn (Hermann et al., 2024). Thus the spurious attribute should be more available to the model when learning its decision rules (Hermann et al., 2024) and given as a condition when we define spurious correlation.

***The marginal distribution of the target attribute should also be accounted for.*** In $Corr_{tcp}$ and $Corr_{scp}$ measure of spurious correlation, the marginal distribution of the target attribute is not taken into account. This is inaccurate for even if the spurious and the target attribute are statistically independent, the value of $Corr_{tcp}$ and $Corr_{scp}$ could be high if the marginal distribution of spurious and target attribute is highly skewed, e.g. long-tail distributed (Zhang et al., 2021; 2023b).

***It's better to use diverge rather than predictivity.*** While $Corr_{sce}$ satisfies the above requirements, it measures the entropy difference between the conditional and marginal distribution of the target attribute, i.e. the predictivity difference. This is still inaccurate for when the entropy of the distributions is the same, the conditional distribution could still be highly diverged from the marginal distribution, thus highly correlated with the spurious attribute. However, using divergence of the distributions accurately measures how the given condition affects the distribution shift of the target attribute.

## 2.3 THE PROPOSED ANALYSIS FRAMEWORK

Given the above requirements that need to be satisfied when measuring spurious correlations, we first propose the following feature-level measure, i.e. bias magnitude.

**Bias Magnitude: spurious attribute conditioned divergence.** We propose a feature-level measure of spurious correlation that measures the KL divergence between the conditional and marginal distribution of the target attribute:

$$\rho_a^* = Corr_{scd} = KL(P(y^t), P(y^t|y^s = a)) \tag{1}$$

where $a$ is the biased feature (or value) in the spurious attribute. The proposed measure satisfies all the requirements above. The above measure only describes the bias of a given feature in the dataset, i.e. feature-level bias. To further describe the bias level of a dataset, i.e. dataset or attribute level bias, we further define the prevalence of bias.

**Bias Prevalence.** Consider a set of biased features whose magnitude of the bias is above a certain threshold $\theta$, i.e. $B = \{a|\rho_a^* > \theta\}$. We define the dataset-level bias by taking not only the number but also the prevalence of the biased features:

$$Prv = \sum_{a \in B} P(y^s = a) \tag{2}$$

Here, we further claim and define the existence of Bias-Neutral (BN) samples, referring to samples that do not hold any biased feature defined in $B$. Bias-Neutral sample is a complement to the previous categorization of samples into Bias-Align (BA) and Bias-Conflict (BC) samples, which is only accurate when all samples in the dataset contain a certain biased feature, assumed by existing synthetic benchmarks. *We elaborate on the categorization of samples in Appendix D.*

*In Appendix E.8, we further show that our framework of measuring dataset biases indeed achieves much stronger correlation with the biased behaviour of models*, i.e., model biases, compared to previous measures (Pearson correlation of 0.98 v.s. 0.78).

## 2.4 OBSERVATION ON REAL-WORLD BIASES

Given the dataset assessing framework proposed above, we are now able to analyze how are dataset biases in existing benchmarks different from that in the real world.

**The magnitude of biases in real-world datasets is low.** As shown in Figure 2(a), the magnitude of biases in real-world datasets is significantly lower than that in existing benchmarks, consistent across various modalities. It is surprising to see how low the magnitude of biases in the real-world dataset is, yet still captured by models (Li & Liu, 2022).

**The prevalence of bias in real-world datasets is low.** As shown in Figure 2(b), the bias prevalence of real-world datasets is also lower than that in existing benchmarks across all thresholds. Considering the bias magnitude of real-world datasets is generally low, it seems fair to set the threshold sufficiently low when calculating the bias prevalence of existing datasets. However, even if we set the threshold to 0.1, the bias prevalence of COCO (Lin et al., 2015) and COMPAS (mat) dataset, i.e. 0.08 and 0.15 respectively, are still significantly lower than that of the existing benchmarks, i.e. 1. *In section 3, we further theoretically show that the above observation is not a mere exception but a manifestation of underlying principles with broader implications.*

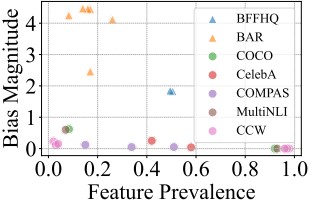

(a) Bias Magnitudes of Datasets

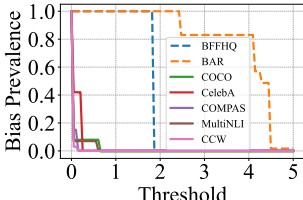

(b) Bias Prevalence of Datasets

Figure 2: With our analysis framework, we can see that the bias magnitude and prevalence of real-world datasets are significantly smaller than that of existing benchmarks.

## 3 THEORETICAL ANALYSIS ON BIASED DISTRIBUTIONS

In this section, we theoretically show that the high bias prevalence (HP) distribution requires two strong assumptions implicitly held by existing benchmarks. Furthermore, the invalidity of the assumptions in real-world scenarios results in low bias prevalence (LP) distributions.

**Data distribution.** Consider a multi-class classification task on the target attribute $y^t \sim \{a_1^t, ...a_n^t\}$ and a spurious attribute $y^s \sim \{a_1^s, ...a_m^s\}$. For any correlated target feature $a_i^t$ and spurious feature $a_j^s$, we have the marginal distribution of the target and spurious feature to be $p_i^t = P(y^t = a_i^t)$ and $p_j^s = P(y^s = a_j^s)$. Then the joint distribution between $y^t$ and $y^s$ can be defined according to the conditional distribution of $y^t$ given $y^s = a_j^s$, i.e. $\tau_j = P(y^t = a_i^t | y^s = a_j^s)$.

**Definition 1** (Simplified Magnitude of Bias). *For the simplicity of theoretical analysis, we propose a simplified version of bias magnitude defined in section 1. Instead of using KL divergence as the measure of distance, we use total variation distance as a proxy for the sake of simplicity:*

$$\rho_j = \tau_j - p_i^t \tag{3}$$

*The simplification is consistent for it satisfies all the conditions proposed in section 2.*

**Definition 2** (Biased Feature). *We consider a feature $y^s = a_j^s$ biased if the ratio of its bias magnitude $\rho_j$ to its theoretical maximum $\rho_j^{max} = 1 - p_i^t$ is above certain threshold $0 \leq \theta \leq 1$:*

$$\phi_j = \frac{\rho_j}{\rho_j^{max}} > \theta \tag{4}$$

**Definition 3** (High Bias Prevalence Distribution). *We consider distribution as a high bias prevalence distribution only if both $y^s = a_j^s$ and $y^s \neq a_j^s$ are biased, i.e. $\phi_j > \theta, \phi_{\neq j} > \theta$.*

Note that the definitions above are adjusted and different from those defined in section 2.3 for the simplicity of the analysis. We then propose the two assumptions implied by high prevalence distributions, whose *proof can be found in Appendix C.*

**Proposition 1** (High bias prevalence distribution assumes matched marginal distributions). *Assume feature $y^s = a_j^s$ is biased. Then high bias prevalence distribution, i.e. feature $y^s \neq a_j^s$ is biased as well, implies that the marginal distribution of $a_i^t$ and $a_j^s$ is matched, i.e. $\lim_{\theta \to 1} p_j^s = p_i^t$.*

**Proposition 2** (High bias prevalence distribution further assumes uniform marginal distributions even if they are matched). *Given that the marginal distribution of $a_j^s$ and $a_i^t$ are matched and not uniform,*

*i.e.* $p = p_i^s = p_j^t < 0.5$. *The bias magnitude of sparse feature, i.e.* $\rho_j^*$, *is monotone decreasing at* $p$, *with* $\lim_{p \to 0^+} \rho_j^* = -log(1 - \phi_j)$. *The bias magnitude of the other features, i.e.* $\rho_{\neq j}^*$, *is monotone increasing at* $p$, *with* $\lim_{p \to 0^+} \rho_{\neq j}^* = 0$.

**Remark 1.** Proposition 2 reveals the fact that as the distribution of attributes becomes increasingly skewed, i.e. $p$ approaches 0, the magnitude of bias for features diverges, the magnitude of feature $a_j^s$ increases while the magnitude of other features $a_{\neq j}^s$ approaches 0. This results in extremely biased single feature and unbiased other features, resulting in LP distributions.

## 4 Methodology

In Section 4.1, we first propose a systematic evaluation framework for real-world debiasing based on our empirical (Section 2) and theoretical (Section 3) insights. Then, in Section 4.2 and 4.3, we dive into how real-world biases pose a challenge to existing methods. Finally, in Section 4.4, we propose a simple-yet-effective approach to adapt existing methods to the real-world scenarios.

### 4.1 Systematic evaluation framework for real-world debiasing

Based on our empirical and theoretical insights, we further introduce two novel types of bias inspired by real-world applications as follows. Together with high magnitude high prevalence (HMHP) distribution in existing benchmarks and other real-world challenges, we form a systematic evaluation framework for real-world debiasing. *Please refer to Appendix B for full description of the framework.*

**Low Magnitude Low Prevalence (LMLP) Bias.** Inspired by the distribution of the COMPAS (mat) dataset shown in Figure 2(a), bias in the real world might be low in both magnitude and prevalence. To take it even further, we should not even assume the dataset is biased at all when applying debiasing methods, because we usually lack such information in practice. Thus, unbiased data distribution can be considered as a special case of the distribution.

**High Magnitude Low Prevalence (HMLP) Bias.** As shown in Figure 2, the COCO (Lin et al., 2015) dataset may contain features with relatively high bias magnitude, yet low bias prevalence in the dataset due to the sparsity of the biased feature, i.e. low feature prevalence.

### 4.2 Existing paradigm for debiasing w/o bias label

In recent years, research in the field of debiasing has been more focused on the practical setting of debiasing w/o bias supervision. Though different in technical details, they generally adopt *a biased auxiliary model to capture the bias*, followed by techniques to learn a debiased model with the captured bias. The bias capture process is based on the assumption that the spurious attributes are easier and learned more preferentially than the target attribute, thus bias could be captured by an auxiliary model $M_b$ w/o bias labels. To utilize $M_b$ for debiasing, the generally shared heuristic is that BC samples should be relatively difficult for $M_b$ but not the debiased model $M_d$. One widely adopted implementation of the heuristic is the loss-based sample reweighing scheme $W(x)$ proposed by Nam et al. (2020), which we use for our analysis:

$$W(x) = \frac{CE(M_b(x), y)}{CE(M_d(x), y) + CE(M_b(x), y)} \tag{5}$$

where $(x, y)$ are samples from the training data and $CE(\cdot, \cdot)$ is the cross entropy loss. *Please refer to Appendix F for detailed review on existing debiasing methods w/o bias label.*

### 4.3 The Sparse bias capturing challenge in real-world debiasing

We claim that the effectiveness of the critical bias capture module in existing methods relies on the HP assumption of existing benchmarks, which does not generalize to LP biases in the real world. It is assumed that the biased model $M_b$ predicts according to the bias within the training data Sreelatha et al. (2024); Han et al. (2024), giving high loss to BC samples and low loss to BA samples (Zhao et al., 2023; Lee et al., 2023). Existing works attribute this loss difference to the fact that spurious attributes are easier (Nam et al., 2020; Lim et al., 2023), i.e., learn more preferentially by models,

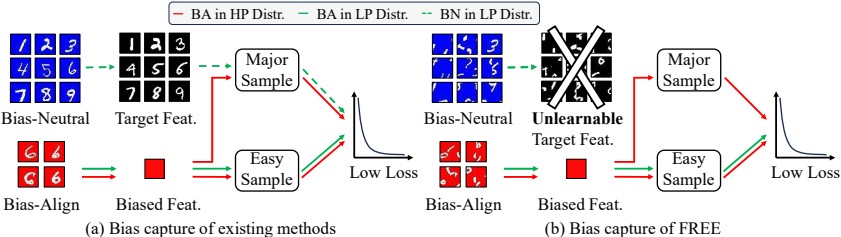

Figure 3: The bias capture process of biased models on LP and HP datasets. Assuming the red background is spuriously correlated with digit 6, and only the major learning of the biased models is illustrated with arrows. DiD eliminates the undesired learning of BN samples on the LP dataset in Figure 3(a) by destroying the target feature, as shown in Figure 3(b).

making BA the *easy sample*. While such a claim is true, the dominance of BA samples in the HP datasets is another vital contributing factor to the loss difference, for *dominant/major samples* are learned more frequently than others, as shown in Figure 3(a). **The existing assumption of dense bias (BA samples are dominant) makes the bias capture process much less challenging.**

However, for LP biases in the real world, while BA samples are still easier to learn due to the biased feature, the dominant/major samples in the training data are no longer BA samples, but rather BN samples. This not only results in the loss difference between BA and BC samples decreasing but also causes low loss on BN samples, as shown in Figure 3(a). According to sample weighing scheme 5, such low loss on BN samples further leads to low weights for BN samples when training the debiased model, which is unintended as BN samples carry an abundant amount of knowledge concerning the target attribute without the interference of the spurious features. **In other words, the sparsity of real-world biases makes accurate bias capturing much more challenging, leading to severe degradation in the subsequent debiasing process.** We empirically prove our claim in section 5.

### 4.4 BIAS CAPTURE WITH FEATURE DESTRUCTION

Based on our analysis in section 4.3, we introduce a simple yet effective enhancement to the bias capture module in the existing framework. We name the refined framework as Debias in Destruction (DiD). As shown in Figure 3(b), the problem with the existing bias capture method comes from the side branch learning on BN samples of the biased auxiliary model, which not only captures the bias but also learns the target feature. This is undesired for this further causes the overlooking of BN samples when training the debiased model, as discussed in section 4.3.

To prune the side branch learning of the target features, it is intuitive to destroy the target feature and make them unlearnable when training the biased model, as shown in Figure 3(b). Such action is practical because the target features we intend to learn are usually clear, and no information on the biased feature is required. Specifically, we can achieve this by applying target feature destructive data transformation when training the biased model:

$$Loss_b^{DiD} = Loss_b(M_b(T_{fd}(x)), y)$$

where $Loss_b$ is the original loss for training the biased model (e.g. $CE$ and $GCE$ (Zhang & Sabuncu, 2018)), and $T_{fd}(\cdot)$ is the feature destruction transformation. As an example, in visual recognition tasks, the shape of objects is a basic element of human visual perception (Geirhos et al., 2019). Therefore, the patch-shuffle destruction of shape (Lee et al., 2024) when capturing bias from visual recognition datasets is a feasible approach. As for NLP tasks, we adopt a word shuffling approach which we will elaborate on in Appendix D.5.

## 5 EXPERIMENTS

In this section, with the systematic evaluation framework proposed in Section 4.1, we empirically examine our findings on Sparse bias capturing challenge and the proposed approach, along with additional results on real-world debiasing.

## 5.1 EXPERIMENTAL SETTINGS

**Metrics.** Following previous works, we adopt the accuracy of BC samples (BC), the average accuracy on the balanced test set (**Avg**), and the worst group accuracy (Worst Acc.). **Datasets.** We adopt *8 datasets* in various modalities for evaluation. Specifically, we adopt the basic setting of Colored MNIST and Corrupted CIFAR10 to implement the distributions within the proposed systematic evaluation framework. We also evaluated our method on more existing synthetic benchmarks who is more complex in terms of the target and spurious feature: BAR, NICO, and WaterBirds. We also adopt 2 real-world NLP datasets MultiNLI and CivilComments-WILDS, 3 real-world tabular datasets COMPAS, Adult, German, and 1 real-world image dataset CelebA. **Baselines.** We adopt *9 baselines*, covering classic and recently proposed methods. ERM directly applies standard training on the biased datasets. LfF (Nam et al., 2020) is a pioneer work to debias w/o bias label. DisEnt (Lee et al., 2021) disentangles bias and intrinsic features and applies feature augmentation when training the debiased model. BEL, BED (Lee et al., 2023), DPR Han et al. (2024), DeNetDM Sreelatha et al. (2024) are recently proposed methods. JTT (Liu et al., 2021) is a classic method adapted to both the image and NLP domains. Group DRO (Sagawa* et al., 2020) requires bias supervision used as an upper bound. *Detailed discription in Appendix D.*

## 5.2 MAIN RESULTS

**Existing bias capturing degrade on LP biases, while DiD significantly boost the performance.**As shown in Table 1, while performing decently on HMHP distributed datasets, existing methods degrade on both LMLP and HMLP biases, with BC and Avg accuracy both lower than the ERM baseline. This indicates the degradation of existing bias capture module on LP biases, sacrificing utility (Avg), without improving worst group performance (BC). We further tested the effectiveness of DiD by combining DiD with existing methods. As shown in Table 1, when combined with DiD, the BC and Avg accuracy both improve on existing HMHP benchmarks. On LMLP and HMLP datasets, the superiority of DiD is even more prominent, where BC and average accuracy both improve significantly, achieving an average of +16.6 and +11.7 for LfF and DisEnt, respectively. *Results on more existing methods in Appendix E.2 further show the generality of our findings.*

Table 1: The performance of our approach is presented in *absolute accuracy increase* of existing methods. Results show that existing debiasing methods perform poorly on LP distributions, yet our method effectively boosts the performance of existing methods across all types of biases.

| | Colored MNIST | | | | | | Corrupted CIFAR10 | | | | | |
| | LMLP | | HMLP | | HMHP | | LMLP | | HMLP | | HMHP | |
| Algorithm | BC | Avg | BC | Avg | BC | Avg | BC | Avg | BC | Avg | BC | Avg |
|---|---|---|---|---|---|---|---|---|---|---|---|---|
| ERM | 91.1 | 91.7 | 85.2 | 89.8 | 48.5 | 53.4 | 62.5 | 64.3 | 55.9 | 65.1 | 29.4 | 35.4 |
| LfF | 68.4 | 69.7 | 58.0 | 63.3 | 65.6 | 64.6 | 55.0 | 55.4 | 47.7 | 54.1 | 35.3 | 39.0 |
| + DiD | **+22.6** | **+21.4** | **+32.6** | **+25.8** | **+1.3** | **+3.4** | **+7.0** | **+7.3** | **+7.1** | **+8.9** | **+1.8** | **+2.5** |
| DisEnt | 73.9 | 74.9 | 66.5 | 72.2 | 68.3 | 67.4 | 55.5 | 56.1 | 52.5 | 54.5 | 36.0 | 39.5 |
| + DiD | **+17.2** | **+16.5** | **+22.0** | **+16.8** | **+0.8** | **+3.1** | **+5.4** | **+5.9** | **+2.8** | **+7.1** | **+3.0** | **+3.3** |
| BEL | 83.6 | 83.5 | 80.0 | 82.3 | 66.9 | 67.6 | 52.1 | 54.0 | 51.0 | 54.0 | 31.5 | 36.6 |
| + DiD | **+5.7** | **+6.1** | **+9.1** | **+4.9** | -0.5 | **+0.7** | **+1.1** | **+0.2** | -0.8 | **+0.1** | **+1.4** | **+0.8** |
| BED | 81.1 | 81.0 | 77.6 | 80.2 | 67.5 | 68.5 | 56.6 | 57.2 | 49.1 | 56.3 | 34.2 | 38.6 |
| + DiD | **+8.7** | **+9.0** | **+11.7** | **+5.5** | **+2.0** | **+2.5** | **+4.3** | **+4.2** | **+4.9** | **+5.1** | **+3.5** | **+3.2** |

**DiD is consistently effective on complex visual features.**As shown in Table 2, our approach is not merely effective under the setting of Colored MNIST and Corrupted CIFAR10, but rather consistently effective on datasets with more complex sets of target and spurious features. This shows the adaptability of DiD to more sophisticated visual data. Refer to Appendix D.2 for the metrics used.

**DiD is effective on real-world datasets in various modalities.** We choose JTT as the baseline for this part of the experiment for it is a classic method adopted to both the image and NLP domain. As shown in Table 7, our approach is consistently effective on real-world datasets in various modalities,

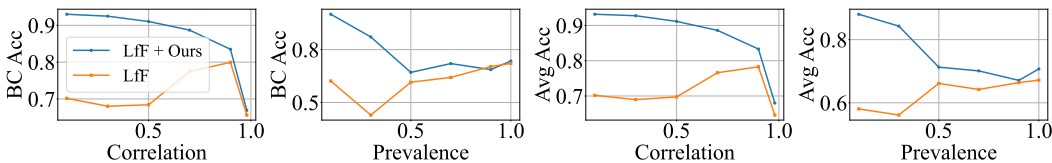

Figure 4: The performance of debiasing methods under various bias magnitudes and prevalence.

further demonstrating its generalizability.We report the results on 3 real-world tabular datasets in Appendix E.9.

Table 2: Results on 3 datasets with more complex and realistic sets of features further show the effectiveness of our approach.

| Algorithm | BAR | NICO | WaterBirds |
|---|---|---|---|
| ERM | 35.32 $_{\pm 0.27}$ | 42.61 $_{\pm 0.33}$ | 56.53 $_{\pm 0.27}$ |
| LfF | 37.73 $_{\pm 1.00}$ | 51.69 $_{\pm 3.06}$ | 50.02 $_{\pm 0.00}$ |
| + DiD | **+3.34** $_{\pm 1.69}$ | **+2.80** $_{\pm 3.10}$ | **+4.47** $_{\pm 0.13}$ |
| DisEnt | 59.11 $_{\pm 1.75}$ | 39.73 $_{\pm 0.58}$ | 56.75 $_{\pm 4.19}$ |
| + DiD | **+3.92** $_{\pm 0.62}$ | **+16.55** $_{\pm 1.29}$ | **+11.29** $_{\pm 0.69}$ |
| BEL | 38.40 $_{\pm 0.65}$ | 44.09 $_{\pm 1.95}$ | 52.98 $_{\pm 0.28}$ |
| + DiD | **+1.08** $_{\pm 1.67}$ | **+8.23** $_{\pm 1.49}$ | **+1.92** $_{\pm 0.12}$ |
| BED | 62.74 $_{\pm 1.23}$ | 39.58 $_{\pm 0.91}$ | 53.85 $_{\pm 2.14}$ |
| + DiD | **+0.70** $_{\pm 1.20}$ | **+13.50** $_{\pm 1.62}$ | **+1.99** $_{\pm 3.65}$ |

Table 3: We experiment with three feature destruction methods with various hyperparameters on HMLP distributed dataset with LfF.

| $T_{fd}$ | param | BC | Avg |
|---|---|---|---|
| N/A | N/A | 47.70 $_{\pm 3.58}$ | 54.15 $_{\pm 3.02}$ |
| pixel-shuffle | 1 | 51.44 $_{\pm 1.01}$ | 55.43 $_{\pm 0.20}$ |
| patch-shuffle | 2 | 51.07 $_{\pm 0.48}$ | 55.29 $_{\pm 0.27}$ |
| | 4 | 49.41 $_{\pm 0.26}$ | 55.40 $_{\pm 0.26}$ |
| | 8 | 54.81 $_{\pm 0.74}$ | 63.06 $_{\pm 0.77}$ |
| | 16 | 49.74 $_{\pm 1.10}$ | 53.69 $_{\pm 0.31}$ |
| center-occlusion | 8 | 45.19 $_{\pm 1.41}$ | 51.61 $_{\pm 1.31}$ |
| | 16 | 47.26 $_{\pm 0.54}$ | 50.94 $_{\pm 0.59}$ |
| | 24 | 49.00 $_{\pm 0.80}$ | 52.60 $_{\pm 0.55}$ |
| | 32 | 52.44 $_{\pm 0.87}$ | 55.76 $_{\pm 0.16}$ |

## 5.3 ANALYSIS

**Accuracy of bias capturing.** We further examine the accuracy of bias capturing by tracking the weights of samples to see if they align with our hypothesis in Section 4. Figure 7(a) and 7(b) plots the average weights of all kinds of samples on HMLP biases, which shows that the degradation of existing methods is indeed caused by the *Sparse bias capturing* challenge in section 4.3, overlooking BN samples when training the debiased model $M_d$. Figure 7(c) and 7(d) track the sample weight of BN samples. As we can see, DiD significantly raise the weights of the BN sample, which demonstrates more accurate bias capturing and the effectiveness of our design.

**Ablation on destruction methods.** As shown in Table 3, we examine three feature destruction methods: pixel-shuffling, patch-shuffling, and center occlusion. We observed that patch-shuffle with patch-size 8 exhibits the best performance on Corrupted CIFAR10 of size 32x32.

**Effect of bias magnitude and prevalence in debiasing.** As shown in Figure 4, we use the correlation $Corr_{scp}$ defined in section 2 as a proxy for the bias magnitude and vary it from low to high. With the increase of the bias magnitude, the performance of LfF first increases as the data becomes biased, and then decreases as the bias magnitude becomes extremely high. As shown in 4, we vary the prevalence of bias by controlling the number of biased features. With the increase of the bias prevalence, the performance generally keeps increasing for its reliance on high prevalence as discussed in Section 4. In all cases DiD consistently improves the performance across the spectrum.

## 5.4 ADDITIONAL STUDIES ON REAL-WORLD DEBIASING

We explore additional questions in real-world debiasing on the systematic evaluation framework: **1. How do debiasing methods perform on unbiased datasets? (Appendix E.3) 2. How effective is DiD on multi-bias scenarios? (Appendix E.4) 3. Is DiD effective on bias detection tasks as well? (Appendix E.5) 4. And more (Appendix E.1, E.6, E.7, E.8).**

## 6 CONCLUSIONS AND DISCUSSION

In this work, we revisit the task of debiasing under real-world scenarios. Through solid empirical and theoretical analysis, we found a noticeable gap between existing evaluations and real-world requirements. We further fill the gap with a systematic evaluation framework for real-world debiasing. We also uncover a novel challenge in real-world debiasing, along with a simple yet effective method to address it. In Appendix G, we further discuss the limitations and future directions of this work.

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

## A  MORE VISUALIZATIONS OF BIASED DISTRIBUTIONS

We plot the biased distributions of more existing benchmarks as follows:

**WaterBirds.**  WaterBirds Liu et al. (2021) is a synthetic dataset with the task of classify images of birds as "waterbird" and "landbird", which is adopted as a benchmark for debiasing methods. The label of WaterBirds is spuriously correlated with the image background, i.e. Place attribute, which is either "land" or "water". The joint distribution between the Place and Bird attribute of the WaterBirds dataset is plotted in Figure 5a.

Additional visualization of the biased distribution within real-world datasets is also plotted as follows:

**CelebA.**  CelebA Liu et al. (2015) is a dataset for face recognition where each sample is labeled with 40 attributes, which has been adopted as a benchmark for debiasing methods. Following the experiment configuration suggested by Nam et al. [32], we focus on HeavyMakeup attributes that are spuriously correlated with Gender attributes, i.e., most of the CelebA images with heavy makeup are women. As a result, the biased model suffers from performance degradation when predicting males with heavy makeup and females without heavy makeup. Therefore, we use Heavy_Makeup as the target attribute and Male as a spurious attribute. The joint distribution between the Male and Heavy_Makeup attribute of the CelebA dataset is plotted in Figure 5b. It is clear that the biased distribution of CelebbA aligns with that in other existing benchmarks, forming a "diagonal distribution".

**Adult.**  The Adult Becker & Kohavi (1996) dataset, also known as the "Census Income" dataset, is widely used for tasks such as income prediction and fairness analysis. Each sample is labeled with demographic and income-related attributes. The dataset has been adopted as a benchmark for debiasing methods, particularly focusing on the correlation between race and income. The joint distribution between Race and Income attributes of the Adult dataset is plotted in Figure 5c. It is clear that the biased distribution of Adult does not align with that of other existing benchmarks.

**German.**  The German Hofmann (1994) dataset, also known as the "German Credit" dataset, is commonly used for credit risk analysis and fairness studies. Each sample is labeled with various attributes related to creditworthiness. The dataset serves as a benchmark for debiasing methods, emphasizing the correlation between age and creditworthiness. The joint distribution between Age and Creditworthiness attributes of the German dataset is plotted in Figure 5d. It is clear that the biased distribution of German does not align with that of other existing benchmarks.

**MultiNLI.**  In the NLP domain, the MultiNLI (Williams et al., 2018) dataset, used for natural language inference, shows a strong bias related to negation. As plotted in Figure 5e, the presence of negation words ("has negation") is spuriously correlated with the target labels. For example, sentences containing negation are highly unlikely to have a "neutral" relationship (a joint probability of 0.0074), creating a shortcut for models. It is clear that the biased distribution of MultiNLI does not align with that of other existing benchmarks.

**CivilComments-WILDS.**  The CivilComments-WILDS (CCW) (Koh et al., 2021), a dataset for toxicity detection, contains biases related to identity terms. Figure 5f visualizes the joint distribution of comment toxicity and the mention of racial identities. The dataset is overwhelmingly composed of non-toxic comments associated with "not white" identities (0.86). Furthermore, toxic comments are more frequently associated with "not white" identities (0.1) than "white" identities (0.0098), posing a significant challenge for building fair models. It is clear that the biased distribution of CivilComments-WILDS does not align with that of other existing benchmarks.

**NIH.**  The NIH ChestX-ray dataset, a common benchmark for medical image analysis, also demonstrates significant bias. As shown in Figure A(g), there is a powerful spurious correlation between the target label (Y) and a spurious attribute (A). The vast majority of the dataset consists of samples where Y=0 and A=0 (a joint probability of 0.91), while all other combinations are rare. This imbalance can lead models to rely on attribute A as a shortcut for predicting Y=0, failing in real-world scenarios

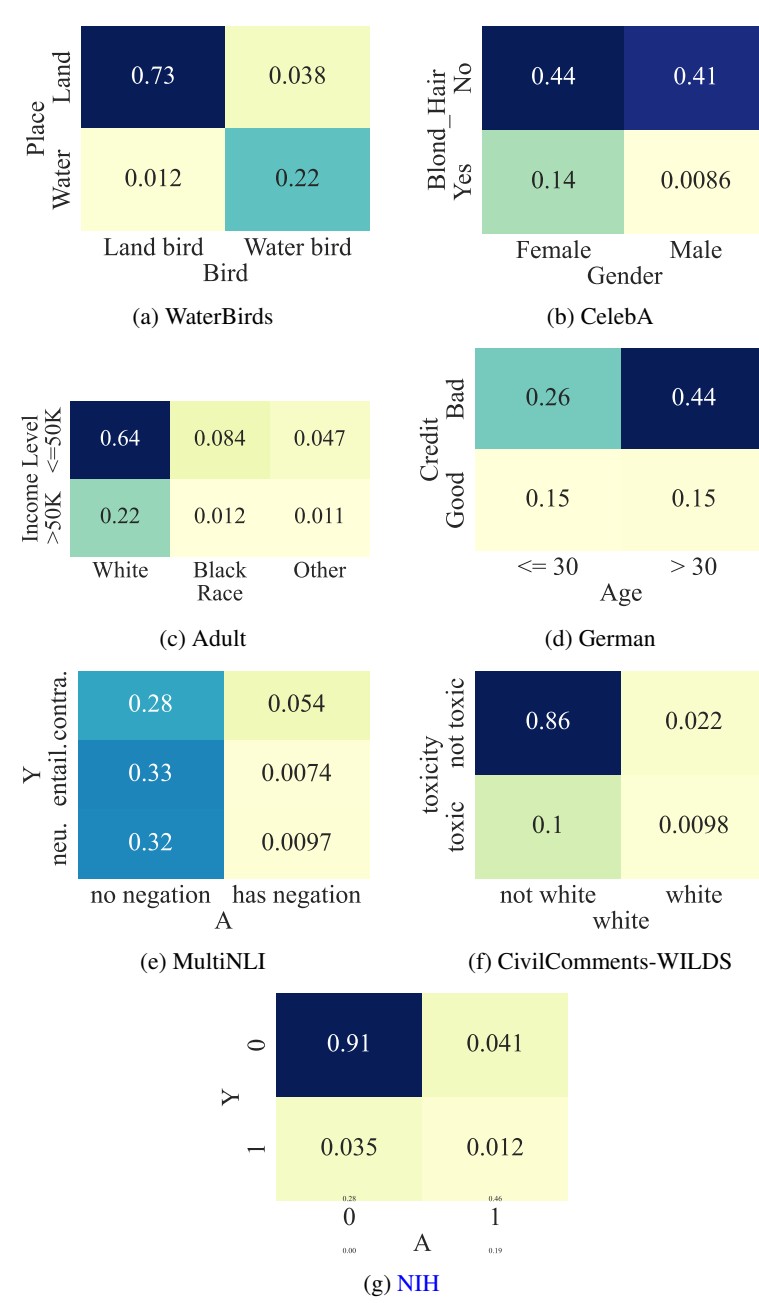

Figure 5: Visualization of the joint distribution for datasets, where the y-axis is the target attribute and the x-axis is the spurious attribute. Figure 5(a) visualize the distribution of existing benchmarks. Figure 5(b), 5(c), 5(d), 5(e), 5(f), and 5(g) visualize the distribution of real-world datasets. The biased distribution of existing benchmarks and real-world datasets is not alike.

where this correlation does not hold. It is clear that the biased distribution of NIH does not align with that of other existing benchmarks.

The description of MultiNLI and CivilComments-WILDS can be found in Appendix E.

Table 4: Configurations for biased distributions within the proposed evaluation framework

| Distribution | $|y^t|$ | $|B|$ | $corr_i$ |
|---|---|---|---|
| LMLP | 10 | 10 | 0.5 |
| LMLP' | 10 | 5 | 0.5 |
| HMLP | 10 | 1 | 0.98 |
| HMHP | 10 | 10 | 0.98 |
| Unbiased | 10 | 0 | 0.1 |

## B FINE-GRAINED EVALUATION FRAMEWORK

In this section, we elaborate on the proposed evaluation framework for real-world debiasing. The framework is mainly composed of three parts: Evaluation on various biased distributions in the real world, evaluation on multi-bias scenarios in the real world, and evaluation on other existing benchmarks. **Covering all those aspects, we aim to provide a comprehensive and easy-to-use base for future development in the debiasing field, toward debiasing methods for real-world scenarios**. Code available at https://github.com

### B.1 EVALUATION ON VARIOUS BIASED DISTRIBUTIONS IN THE REAL WORLD

We mathematically and visually demonstrate the biased distribution currently included in the evaluation framework.

Assume a set of biased features $a_i^s \in B$ whose correlated class in the target attribute is defined by a function $g : y^s \to y^t$, which is an injection from the spurious to the target attribute. The bias magnitude of each biased feature is controlled by $corr_i = P(y^t = g(a_i^s)|y^s = a_i^s)$. Then, the empirical distribution of the biased train distribution satisfies the following equations.

For samples with biased feature $a_i^s$ within $B$:

$$P(y^s = a_i^s, y^t = a^t) = \begin{cases} P(y^s = a_i^s) * corr_i & \text{if } g(a_i^s) = a^t, \\ \frac{P(y^s=a_i^s)*(1-corr_i)}{|y^t|-1} & \text{otherwise}, \end{cases}$$

For samples without biased features and a set of correlated classes $C = \{g(a_i^s) : a_i^s \in B\}$:

$$P(y^s = a^s, y^t = a^t) = \frac{P(y^t = a^t) - \sum_{a_i^s \in B} P(y^s = a_i^s, y^t = a^t)}{|y^s| - |B|}$$

Following the above equations, we further designed LMLP, HMLP, and HMHP biased distributions with the configurations in Table 4. The visualizations of the distributions when the target is a ten-class attribute are in Figure 6.

We note that each biased distributions are not merely the description of datasets used in this work, but rather serves as a general guide used to synthesize or sample biased datasets that reflect biases in the real world.

### B.2 EVALUATION ON MULTI-BIAS SCENARIOS IN THE REAL WORLD

The existence of multiple biases is another challenge in debiasing in the real world Li et al. (2023). We further propose to combine multiple biases with different magnitudes and prevalence (e.g. HMLP + LMLP) together to mimic the complexity of biases in the real world. For instance, based on Corrupted CIFAR10 benchmark, containing 10 target features and 20 spurious features, we can construct a biased dataset with multiple biases of various types. Please refer to Appendix E.4 for an example of this setting.

### B.3 EVALUATION ON OTHER EXISTING BENCHMARKS

We also ensemble other popular benchmarks in the field of debiasing in an easy-to-use fashion to facilitate future research. Please refer to Appendix D.2 for details.

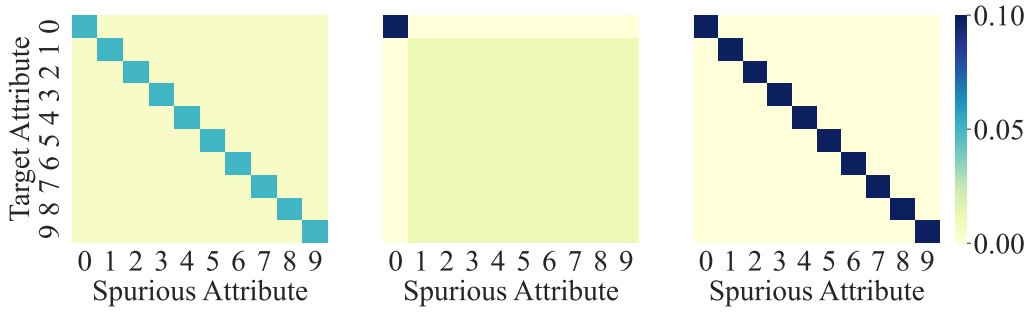

Figure 6: Visualization of biased distributions within the proposed evaluation framework under ten-class classification task. The left, middle, and right plots are visualizations for LMLP, HMLP, and HMHP distribution respectively.

## C  THEORETICAL PROOFS

### C.1  PRELIMINARY

Consider a classification task on the target attribute $y^t \sim \{a_1^t, ... a_n^t\}$ and a spurious attribute $y^s \sim \{a_1^s, ... a_m^s\}$. For any correlated target $a_i^t$ and spurious feature $a_j^s$, we have the marginal distribution of the target and spurious feature to be $p_i^t = P(y^t = a_i^t)$ and $p_j^s = P(y^s = a_j^s)$. Then the joint distribution between $y^t$ and $y^s$ can be defined according to the conditional distribution of $y^t$ given $y^s = a_j^s$, i.e. $\tau_j = P(y^t = a_i^t | y^s = a_j^s)$. Specifically, we can derive the probability of each subgroup in the distribution:

$$P(y^s = a_j^s, y^t = a_i^t) = p_j^s \cdot \tau_j, \tag{6}$$

$$P(y^s = a_j^s, y^t \neq a_i^t) = p_j^s (1 - \tau_j), \tag{7}$$

$$P(y^s = a_j^s, y^t \neq a_i^t) = p_i^t - p_j^s \cdot \tau_j, \tag{8}$$

$$P(y^s \neq a_j^s, y^t \neq a_i^t) = 1 - p_i^t - p_j^s (1 - \tau_j) \tag{9}$$

Furthermore, as feature $y^s = a_j^s$ and $y^t = a_i^t$ are correlated, i.e. $\tau_j > p_i^t$, the complement case of $y^s \neq a_j^s$ and $y^t \neq a_i^t$ is also bound to be correlated, treated as complement features.

### C.2  PROOF OF PROPOSITION 1

Proposition 1 shows that high bias prevalence distribution assumes matched marginal distributions.

**Proposition 1.** *Assume feature $y^s = a_j^s$ is biased. Then high bias prevalence distribution, i.e. feature $y^s \neq a_j^s$ is biased as well, implies that the marginal distribution of $a_i^t$ and $a_j^s$ is matched, i.e. $\lim_{\theta \to 1} p_j^s = p_i^t$.*

*Proof.* We first derive the upper and lower bound of the $p_j^s$, and then we can prove the proposition with the squeeze theorem Stewart (2012).

According to the condition that both features in the spurious attribute are biased and the definition of biased feature in ref, we can have the following inequalities:

$$\rho_j > \theta \cdot \rho_j^{max} = \theta \cdot (1 - p_i^t), \tag{10}$$

$$\rho_- > \theta \cdot \rho_{\neq j}^{max} = \theta \cdot p_i^t \tag{11}$$

where $0 < \theta \leq 1$ is the threshold.

We can also derive the simplified bias magnitude of feature $y^s \neq a_j^s$ based on the conditional distribution, and find its relationship with $\rho_j$:

$$\rho_- = \tau_{\neq j} - p^t_- \tag{12}$$

$$= \frac{1 - p^t_i - p^s_j(1 - \tau_j)}{1 - p^s_j} - (1 - p^t_i) \tag{13}$$

$$= \frac{p^s_j(\tau_j - p^t_i)}{1 - p^s_j} \tag{14}$$

$$= \frac{p^s_j}{1 - p^s_j}\rho_j \tag{15}$$

We can then derive the lower bound of $p^s_j$ with the above equation and inequalities:

$$\frac{p^s_j}{1 - p^s_j}(1 - p^t_i) \geq \frac{p^s_j}{1 - p^s_j}\rho_j = \rho_- \geq \theta \cdot p^t_i \tag{16}$$

$$p^s_j \geq \frac{\theta \cdot p^t_i}{1 - p^t_i + \theta \cdot p^t_i} \geq \theta \cdot p^t_i = LB(\theta) \tag{17}$$

We can also derive the following equation and inequalities of $\tau_j$ according to its definition.

$$\tau_j = \frac{p^s_j \cdot P(y^s = a^s_j | y^t = a^t_i)}{p^s_j} \leq \frac{p^t_i}{p^s_j} \tag{18}$$

$$\tau_j = p^t_i + \rho_j \geq \theta(1 - p^t_i) + p^t_i \tag{19}$$

Then we can derive the upper bound of $p^s_j$:

$$\theta(1 - p^t_i) + p^t_i \leq \tau_j \leq \frac{p^t_i}{p^s_j} \tag{20}$$

$$p^s_j \leq \frac{p^t_i}{\theta(1 - p^t_i) + p^t_i} = UB(\theta) \tag{21}$$

We then demonstrate the convergence of the $LB(\theta)$ and $UB(\theta)$ as $\theta \to 1$:

$$\lim_{\theta \to 1} LB(\theta) = \lim_{\theta \to 1} \theta \cdot p^t_i = p^t_i \tag{22}$$

$$\lim_{\theta \to 1} UB(\theta) = \lim_{\theta \to 1} \frac{p^t_i}{\theta(1 - p^t_i) + p^t_i} = p^t_i \tag{23}$$

Finally, we can prove the proposition according to the squeeze theorem Stewart (2012):

$$LB(\theta) \leq p^s_j \leq UB(\theta) \tag{24}$$

$$\lim_{\theta \to 1} p^s_j = \lim_{\theta \to 1} LB(\theta) = \lim_{\theta \to 1} UB(\theta) = p^t_i \tag{25}$$

### C.3 PROOF OF PROPOSITION 2

Proposition 2 shows that high bias prevalence distribution implies uniform marginal distributions.

**Proposition 2.** *Given that the marginal distribution of $a^s_j$ and $a^t_i$ are matched and not uniform, i.e. $p = p^s_i = p^t_j < 0.5$. The bias magnitude of sparse feature, i.e. $\rho^*_j$, is monotone decreasing at $p$, with $\lim_{p \to 0+} \rho^*_j = -log(1 - \phi_j)$. The bias magnitude of the other features, i.e. $\rho^*_{\neq j}$, is monotone increasing at $p$, with $\lim_{p \to 0+} \rho^*_{\neq j} = 0$.*

*Proof.* Given the distribution proposed in section C.1 and the condition $p = p^s_j = p^t_i < 0.5$, we further use $\phi_j = \frac{\rho_j}{\rho^{max}_j}$ to express $\tau$:

$$\tau_j = p + \phi_j(1 - p) \tag{26}$$

$$\tau_{\neq j} = 1 - p + \phi_j \cdot p \tag{27}$$

We can then derive the bias magnitude of the sparse feature $y^s = a_j^s$, given $p = p_j^s = p_i^t < 0.5$, and warp it with a function $t(p)$.

$$\rho_j^* = KL(P(y^t), P(y^t|y^s = a_j^s)) \tag{28}$$

$$= p \cdot log(\frac{p}{\tau_j}) + (1-p) \cdot log(\frac{1-p}{1-\tau_j}) \tag{29}$$

$$= p \cdot log(\frac{p}{p + \phi_j(1-p)}) + (1-p) \cdot log(\frac{1-p}{1-p-\phi_j(1-p)}) \tag{30}$$

$$= p \cdot log(\frac{p}{p + \phi_j(1-p)}) + (1-p) \cdot log(\frac{1}{1-\phi_j}) \tag{31}$$

$$= p \cdot log(\frac{p(1-\phi_j)}{p + \phi_j(1-p)}) + log(\frac{1}{1-\phi_j}) = t(p) \tag{32}$$

We further derive the partial derivative of $\rho_j^*$ on $p$ as follows:

$$\frac{\partial t(p)}{\partial p} = p \cdot log(\frac{p(1-\phi_j)}{p + \phi_j(1-p)}) + 1 - \frac{p(1-\phi_j)}{p + \phi_j(1-p)} \tag{33}$$

Here we apply substitution method to replace $\frac{p(1-\phi_j)}{p+\phi_j(1-p)}$ with $x$:

$$\frac{\partial t(p)}{\partial p} = f(x) = logx - (x-1) \tag{34}$$

$$0 < x = \frac{p(1-\phi_j)}{p + \phi_j(1-p)} \leq 1 \tag{35}$$

We then show that $f(x)$ is monotone increasing in the interval $0 < x \leq 1$ and the critical point is at $x = 1$.

$$f'(x) = \frac{1}{x} - 1 \geq 0 \tag{36}$$

$$f(1) = 0 \tag{37}$$

Thus, we have $f(x) < 0$ in the interval $0 < x \leq 1$, proving $\rho_j^* = t(p)$ to be monotone decreasing at $p$.

$$\frac{\partial \rho_j^*}{\partial p} = \frac{\partial t(p)}{\partial p} < 0 \tag{38}$$

Similarly, we can derive the bias magnitude of the dense feature $y^s \neq a_j^s$, and see that it is just $t(1-p)$

$$\rho_{\neq j}^* = KL(P(y^t), P(y^t|y^s \neq a_j^s)) \tag{39}$$

$$= (1-p) \cdot log(\frac{(1-p)(1-\phi_j)}{1-p+\phi_j \cdot p}) + log(\frac{1}{1-\phi_j}) \tag{40}$$

$$= t(1-p) \tag{41}$$

As a result, we can prove the monotonicity of $\rho_{\neq j}^*$ with the chain rule.

$$\frac{\partial \rho_{\neq j}^*}{\partial p} = \frac{\partial t(1-p)}{\partial p} \tag{42}$$

$$= \frac{\partial t(1-p)}{\partial(1-p)} \cdot \frac{\partial(1-p)}{\partial p} \tag{43}$$

$$= -\frac{\partial t(1-p)}{\partial(1-p)} \tag{44}$$

$$= -\frac{\partial t(p)}{\partial p} > 0 \tag{45}$$

We can then derive the convergence of sparse feature bias magnitude $\rho_j^*$ when $p$ approaches 0 with L'Hôpital's Rule Stewart (2012).

$$\lim_{p \to 0^+} \rho_j^* = \lim_{p \to 0^+} t(p) \tag{46}$$

$$= \lim_{p \to 0^+} (p \cdot log(\frac{p(1-\phi_j)}{p + \phi_j(1-p)})) + log(\frac{1}{1-\phi_j}) \tag{47}$$

$$= \lim_{p \to 0^+} (p \cdot log(p)) + \lim_{p \to 0^+} (p \cdot log(\frac{1-\phi_j}{p + \phi_j(1-p)})) + log(\frac{1}{1-\phi_j}) \tag{48}$$

$$= \lim_{p \to 0^+} \frac{log(p)}{\frac{1}{p}} + log(\frac{1}{1-\phi_j}) \tag{49}$$

$$= \lim_{p \to 0^+} \frac{(log(p))'}{(\frac{1}{p})'} + log(\frac{1}{1-\phi_j}) \tag{50}$$

$$= \lim_{p \to 0^+} \frac{\frac{1}{p}}{-\frac{1}{p^2}} + log(\frac{1}{1-\phi_j}) \tag{51}$$

$$= log(\frac{1}{1-\phi_j}) \tag{52}$$

Similarly, we can derive the convergence of dense feature bias magnitude $\rho_{\neq j}^*$ when $p$ approaches to 0.

$$\lim_{p \to 0^+} \rho_{\neq j}^* = \lim_{p \to 0^+} t(1-p) \tag{53}$$

$$= \lim_{p \to 1^-} (p \cdot log(\frac{p(1-\phi_j)}{p + \phi_j(1-p)})) + log(\frac{1}{1-\phi_j}) \tag{54}$$

$$= log(1-\phi_j) + log(\frac{1}{1-\phi_j}) \tag{55}$$

$$= 0 \tag{56}$$

## D    EXPERIMENT DETAILS

### D.1    EVALUATION METRICS

Following previous works Nam et al. (2020); Lee et al. (2021); Kim et al. (2022); Lim et al. (2023); Zhao et al. (2023); Lee et al. (2023), we use the accuracy of BC samples and the average accuracy on balanced test set as our main metrics. As a complement, we also present the accuracy of BN and BA samples when analyzing the performance of methods. Formally, we categorize samples according to the attributes $(y^s, y^t)$ and a function $g : y^s \to y^t$ that maps the biased features to its correlated class.

$$BA = \{i|y^s[i] \in B, y^t[i] = g(y^s[i])\} \tag{57}$$

$$BC = \{i|y^s[i] \in B, y^t[i] \neq g(y^s[i])\} \tag{58}$$

$$BN = \{i|y^s[i] \notin B\} \tag{59}$$

where $y^s[i]$ and $y^t[i]$ the attribute value of sample $i$, and $B = \{a|\rho_a^* > \theta\}$ is the set of biased features.

### D.2    DATASETS

**Colored MNIST (Reddy et al., 2021).**    We construct the Colored MNIST dataset based on the MNIST Lecun et al. (1998) dataset and set the background color as the bias attribute. Different from Colored MNIST used in previous work that simply correlates each of the 10 digits with a distinct color, where the strength of the correlation is controlled by setting the number of bias-aligned samples to $\{0.95\%, 0.98\%, 0.99\%, 0.995\%\}$, we proposed a more fine-grained generation process that is capable of various biased distributions, including LMLP, HMLP, HMHP. See Appendix B for more details.

**Corrupted CIFAR10 (Nam et al., 2020).** We construct the Corrupted CIFAR10 dataset based on the CIFAR10 Krizhevsky (2009) dataset and set the corruption as the bias attribute. Different from Corrupted CIFAR10 used in previous work that simply correlates each of the 10 objects with a distinct corruption, where the strength of the correlation is controlled by setting the number of bias-aligned samples to {0.95%, 0.98%, 0.99%, 0.995%}, we proposed a more fine-grained generation process that is capable of various biased distributions, including LMLP, HMLP, HMHP. See Appendix B for more details.

**BAR (Nam et al., 2020).** Biased Action Recognition (BAR) is a semi-synthetic dataset deliberately curated to contain spurious correlations between six human action classes and six place attributes. Following Nam et al. (2020), the ratio of bias-conflicting samples in the training set was set to 5%, and the test set consisted of only bias-conflicting samples. We report the accuracy of bias-conflicting samples following Nam et al. (2020).

**NICO (Kim et al., 2022)** NICO is a real-world dataset for simulating out-of-distribution image classification scenarios. Following the setting used by Wang et al. (2021), we use a curated animal subset of NICO that exhibits strong biases (thus still semi-synthetic), which is labeled with 10 object and 10 context classes for evaluating the debiasing methods. The training set consists of 7 context classes per object class and they are long-tailed distributed (e.g., dog images are more frequently coupled with the 'on grass' context than any of the other 6 contexts). The validation and test sets consist of 7 seen context classes and 3 unseen context classes per object class. We verify the ability of debiasing a model from object-context correlations through evaluation on NICO. We report the average accuracy on the test set following Kim et al. (2022).

**WaterBirds (Sagawa* et al., 2020).** The task is to classify images of birds as "waterbird" or "landbird", and the label is spuriously correlated with the image background, which is either "land" or "water". We report the worst group accuracy following Liu et al. (2021).

**MultiNLI (Williams et al., 2018).** Given a pair of sentences, the task is to classify whether the second sentence is entailed by, neutral with, or contradicts the first sentence. We use the spurious attribute from Sagawa* et al. (2020), which is the presence of negation words in the second sentence; due to the artifacts from the data collection process, contradiction examples often include negation words.

**CivilComments-WILDS (Koh et al., 2021).** The task is to classify whether an online comment is toxic or non-toxic, and the label is spuriously correlated with mentions of certain demographic identities (male, female, White, Black, LGBTQ, Muslim, Christian, and other religion). We use the evaluation metric from Koh et al. (2021), which defines 16 overlapping groups (a, toxic) and (a, non-toxic) for each of the above 8 demographic identities a, and report the worst-group performance over these groups.

### D.3 BASELINES

**LfF (Nam et al., 2020).** Learning from Failure (LfF) is a debiasing technique that addresses the issue of models learning from spurious correlations present in biased datasets. The method involves training two neural networks: one biased network that amplifies the bias by focusing on easily learnable spurious correlations, and one debiased network that emphasizes samples the biased network misclassifies. This dual-training scheme enables the debiased network to focus on more meaningful features that generalize better across various datasets.

**DisEnt (Lee et al., 2021) .** The DisEnt method enhances debiasing by using disentangled feature augmentation. It identifies intrinsic and spurious attributes within data and generates new samples by swapping these attributes among the training data. This approach significantly diversifies the training set with bias-conflicting samples, which are crucial for effective debiasing. By training models with these augmented samples, DisEnt achieves better generalization and robustness against biases in various datasets.

**JTT (Liu et al., 2021).** JTT is a classic debiasing method w/o bias label, that has been applied to both the image and NLP domains. JTT identifies challenging examples by training an initial model using standard empirical risk minimization (ERM) and collecting misclassified examples into an error set. The second stage involves re-training the model while upweighting the error set to prioritize examples that the first-stage model struggled with. This approach aims to address performance disparities caused by spurious correlations, leading to better generalization across groups with minimal additional annotation costs.

**BE (Lee et al., 2023).** BiasEnsemble (BE) is a recent advancement in debiasing techniques that emphasizes the importance of amplifying biases to improve the training of debiased models. BE involves pretraining multiple biased models with different initializations to capture diverse visual attributes associated with biases. By filtering out bias-conflicting samples using these pre-trained models, BE constructs a refined bias-amplified dataset for training the biased network. This method ensures the biased model is highly focused on bias attributes, thereby enhancing the overall debiasing performance of the subsequent debiased model.

**DPR (Han et al., 2024).** DPR is another recently proposed debiasing method w/o bias label. DPR rectifies biased models through fine-tuning. They construct a small pivotal subset with a higher proportion of bias-conflicting samples using BCSI, which serves as an effective alternative to an unbiased set. Leveraging this pivotal set, they rectify a biased model through fine-tuning with only a few additional iterations.

**DeNetDM (Sreelatha et al., 2024)** . DeNetDM is another recently proposed debiasing method w/o bias label. utilize a technique inspired by the Product of Experts, where one expert is deeper than the other. They propose a strategy where they train a deep debiased model utilizing the information acquired from both deep (perfectly biased) and shallow (weak debiased) network in the previous phase.

**Group DRO (Sagawa* et al., 2020).** Group DRO is a supervised debiasing method aiming to improve the worst group accuracy. It is commonly used as an upper bound in the worst group accuracy for unsupervised methods.

### D.4 IMPLEMENTATION DETAILS

**Reproducibility.** To ensure the statistical robustness and reproducibility of the result in this work, we repeat each experiment within this work 3 times with consistent random seeds [0, 1, 2]. All results are the average of the three independent runs.

**Architecture.** Following Nam et al. (2020); Lee et al. (2021), we use a multi-layer perceptron (MLP) which consists of three hidden layers for Colored MNIST. For the Corrupted CIFAR10, BAR, NICO, WaterBirds dataset, we train ResNet18 He et al. (2015) with random initialization. For CelebA dataset, we train ResNet50 with random initialization, following Liu et al. (2021). For MultiNLI and CivilComments-WILDS datasets, we use Bert for training, following Liu et al. (2021).

**Training hyper-parameters.** We set the learning rate as 0.001, batch size as 256, momentum as 0.9, and number of steps as 25000. We used the default values of hyper-parameters reported in the original papers for the baseline models.

**Data augmentation.** The image sizes are 28×28 for Colored MNIST and 224×224 for the rest of the datasets. For Colored MNIST, we do not apply additional data augmentation techniques. For Corrupted CIFAR10, we apply random crop and horizontal flip transformations. Also, images are normalized along each channel (3, H, W) with the mean of (0.4914, 0.4822, 0.4465) and standard deviation of (0.2023, 0.1994, 0.2010).

**Training device.** We conducted all experiments on a workstation with an Intel(R) Xeon(R) Gold 5220R CPU at 2.20GHz, 256 G memory, and 4 NVIDIA GeForce RTX 3090 GPUs. Note that only a single GPU is used for a single task.

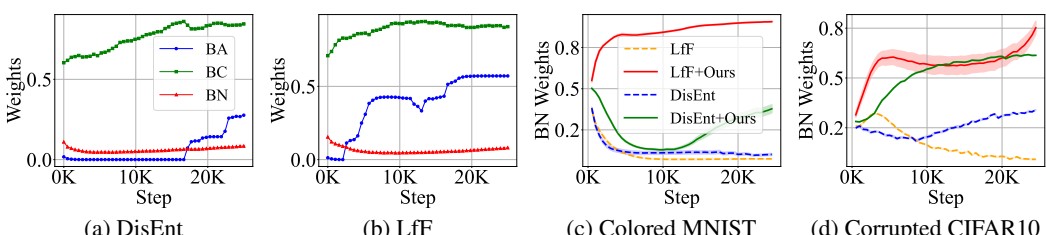

(a) DisEnt      (b) LfF      (c) Colored MNIST      (d) Corrupted CIFAR10

Figure 7: Figure 7(a) and 7(b) support our claim in section 4 that existing debiasing methods tend to overlook BN samples when training on LP distributions. Figure 7(a) and 7(b) show that our approach effectively emphasizes BN samples by raising its weights.

### D.5 DESIGN OF FEATURE DESTRUCTING METHODS

For in visual recognition tasks, the shape of objects is a basic element of human visual perception (Geirhos et al., 2019). Therefore, the patch-shuffle destruction of shape (Lee et al., 2024) when capturing bias from visual recognition datasets is a feasible approach. We adopt the patch-shuffle approach for all the visual dataset within the paper except for CelebA. We apply a gray-scale transformation for CelebA as its recognition task is hair color. Anyhow, the feature destruction method could be highly flexible for different tasks.

For NLP tasks, we first introduce the common biases within the NLP domain followed by a simple design of feature destruction method in the NLP domain. The commonly used NLP datasets for debiasing are MultiNLI and CivilComments-WILDS dataset. Specifically, the bias within the MultiNLI dataset is the correlation between the negation words and the entailment task and the bias within the CivilComments-WILDS dataset is the correlation between words implying demographic identities and the toxicity task. The target features of both datasets are semantic information of the sentences where the position of words matters, and the spurious features are the individual words which is insensitive to positions. Furthermore, such position sensitivity difference between target and spurious features within NLP biases is not limited to these two datasets but rather quite common. For example, CLIP has also been found with the "bag of words" phenomenon (Yuksekgonul et al., 2023), which ignores the semantic meaning of the inputs and relies on words individually for prediction. As a result, a straightforward approach for feature destruction is to shuffle the words within the sentences.

### D.6 APPLYING DiD TO EXISTING METHODS

As aforementioned in the main paper, when applying our method to the existing debiasing methods Nam et al. (2020); Lee et al. (2021; 2023), we do not modify the training procedure of the debiased model $M_d$. For both methods, we train the biased model $M_b$ with target feature destroyed data. This is done by simply adding a feature destructive data transformation during data processing, with minimal computational overhead.

Note, for BE Lee et al. (2023), such feature destructive data transformation is not applied when training the bias-conflicting detectors.

## E ADDITIONAL EMPIRICAL RESULTS

### E.1 DETAILED RESULTS AND EXPLANATIONS OF THE MAIN EXPERIMENTS

The main results in the main paper are presented in the form of performance gain and only contain results of BC accuracy and average accuracy on the unbiased test set, here we present the results in their original form, together with error bars, detailed results of accuracies for BA and BN samples of each dataset as well. Results on the Colored MNIST and Corrupted CIFAR10 datasets can be found in Table 5 and Table 6, respectively. It shows that combining DiD not only boosts the performance of existing debiasing methods but also achieves the best performance.

The performance generally varies between different datasets, different types of biased distribution, and algorithms with and without BiasEnsemble, e.g. between LfF and BEL. Firstly, the inconsistency

between datasets is likely to depend on how thoroughly the target feature is destroyed within the dataset. The target features of Colored MNIST, i.e. digits, are destroyed more completely by patch shuffling, for shape is the only feature within digits. In comparison, the target feature of Corrupted CIFAR10 is more complicated (including shape, texture, color, etc.), and thus can not be thoroughly destroyed by patch shuffling, causing relatively lower performance gain. Secondly, the performance inconsistency between different biased distributions is due to the reliance of existing debiasing methods on the high bias prevalence assumption for bias capturing as discussed in section 4.2. Specifically, as the bias prevalence of the training distribution becomes higher, better bias capture can be achieved even without our method, thus making our improvement on the performance less significant, but still quite effective. This conclusion is supported by our experimental results shown in Figure 5. As for the performance inconsistency between algorithms with and without BiasEnsemble, it is due to the fact that BiasEnsemble is also a method targeted to enhance the bias capture procedure of the debiasing framework. As we can see that BiasEnsemble is much more robust to the change in the bias magnitude and prevalence from Table 1. In other words, certain overlap between the goals of BiasEnsemble and our method resulted in smaller improvement of our method on BiasEnsemble-based baselines.

Table 5: Results on Colored MNIST dataset show that combining DiD not only boosts the performance of existing debiasing methods but also achieves the best performances. The accuracy of BN samples is marked as '-' in LMLP and HMHP distribution for there is no BN sample within the dataset according to our evaluation setting in Appendix D.

| Distr. | Algorithm | Accuracy | | | |
|--------|-----------|----------|---------|--------|---------|
| | | BA acc | BC acc | BN acc | Avg acc |
| | ERM | $97.73 \pm 0.09$ | $91.13 \pm 0.17$ | - | $91.73 \pm 0.16$ |
| | LfF | $80.25 \pm 4.86$ | $68.41 \pm 2.01$ | - | $69.74 \pm 2.41$ |
| | + DiD | $92.16 \pm 0.35$ | $91.03 \pm 0.15$ | - | $91.15 \pm 0.17$ |
| LMLP | BEL | $82.95 \pm 1.68$ | $83.60 \pm 0.85$ | - | $83.53 \pm 0.75$ |
| | + DiD | $93.49 \pm 0.81$ | $89.25 \pm 0.64$ | - | $89.67 \pm 0.54$ |
| | DisEnt | $84.45 \pm 1.72$ | $73.87 \pm 2.52$ | - | $74.93 \pm 2.44$ |
| | + DiD | $\mathbf{94.03} \pm \mathbf{0.66}$ | $\mathbf{91.09} \pm \mathbf{0.24}$ | - | $\mathbf{91.38} \pm \mathbf{0.28}$ |
| | BED | $80.18 \pm 1.94$ | $81.07 \pm 2.50$ | - | $80.98 \pm 2.29$ |
| | + DiD | $91.89 \pm 0.26$ | $89.80 \pm 0.97$ | - | $90.01 \pm 0.89$ |
| | ERM | $99.32 \pm 0.34$ | $85.25 \pm 1.62$ | $90.30 \pm 0.56$ | $89.82 \pm 0.70$ |
| | LfF | $87.76 \pm 4.12$ | $57.98 \pm 3.58$ | $63.72 \pm 3.22$ | $63.35 \pm 3.02$ |
| | + DiD | $82.99 \pm 5.08$ | $\mathbf{90.54} \pm \mathbf{0.74}$ | $\mathbf{89.04} \pm \mathbf{0.84}$ | $89.12 \pm 0.77$ |
| HMLP | BEL | $57.65 \pm 32.14$ | $80.02 \pm 1.10$ | $82.84 \pm 1.68$ | $82.33 \pm 1.93$ |
| | + DiD | $63.95 \pm 15.64$ | $89.11 \pm 1.29$ | $87.28 \pm 1.54$ | $87.22 \pm 1.58$ |
| | DisEnt | $77.55 \pm 7.93$ | $66.52 \pm 8.75$ | $72.69 \pm 5.91$ | $72.18 \pm 6.05$ |
| | + DiD | $\mathbf{88.78} \pm \mathbf{7.24}$ | $88.52 \pm 1.47$ | $89.04 \pm 1.13$ | $\mathbf{88.99} \pm \mathbf{1.16}$ |
| | BED | $41.84 \pm 6.21$ | $77.59 \pm 0.69$ | $80.87 \pm 1.78$ | $80.19 \pm 1.71$ |
| | + DiD | $31.97 \pm 7.08$ | $89.33 \pm 1.07$ | $85.88 \pm 0.86$ | $85.66 \pm 0.89$ |
| | ERM | $99.57 \pm 0.07$ | $48.54 \pm 1.22$ | - | $53.38 \pm 1.10$ |
| | LfF | $57.16 \pm 8.27$ | $65.62 \pm 2.87$ | - | $64.59 \pm 3.31$ |
| | + DiD | $77.84 \pm 2.49$ | $66.91 \pm 1.73$ | - | $68.00 \pm 1.80$ |
| HMHP | BEL | $73.61 \pm 1.03$ | $66.90 \pm 0.43$ | - | $67.57 \pm 0.47$ |
| | + DiD | $\mathbf{85.65} \pm \mathbf{2.53}$ | $66.37 \pm 2.54$ | - | $68.30 \pm 2.50$ |
| | DisEnt | $59.89 \pm 4.19$ | $68.29 \pm 1.43$ | - | $67.45 \pm 1.28$ |
| | + DiD | $83.65 \pm 0.13$ | $69.05 \pm 0.38$ | - | $70.51 \pm 0.33$ |
| | BED | $77.74 \pm 2.51$ | $67.51 \pm 1.33$ | - | $68.53 \pm 1.45$ |
| | + DiD | $84.62 \pm 1.16$ | $\mathbf{69.50} \pm \mathbf{1.23}$ | - | $\mathbf{71.01} \pm \mathbf{1.08}$ |

Table 6: Results on Corrupted CIFAR10 dataset show that combining DiD not only boosts the performance of existing debiasing methods but also achieves the best performances. The accuracy of BN samples is marked as '-' in LMLP and HMHP distribution for there is no BN sample within the dataset according to our evaluation setting in Appendix D.

| Distr. | Algorithm | Accuracy | | | |
|---|---|---|---|---|---|
| | | BA acc | BC acc | BN acc | Avg acc |
| LMLP | ERM | 80.40 ± 0.81 | 62.50 ± 0.15 | - | 64.29 ± 0.06 |
| | LfF | 59.13 ± 0.68 | 55.03 ± 0.04 | - | 55.44 ± 0.09 |
| | + DiD | 69.47 ± 0.96 | **62.04 ± 0.21** | - | **62.78 ± 0.10** |
| | BEL | 70.87 ± 1.30 | 52.10 ± 0.30 | - | 53.98 ± 0.40 |
| | + DiD | 63.23 ± 2.10 | 53.21 ± 0.20 | - | 54.21 ± 0.38 |
| | DisEnt | 61.58 ± 0.57 | 55.45 ± 0.23 | - | 56.06 ± 0.17 |
| | + DiD | **72.23 ± 0.74** | 60.84 ± 0.40 | - | 61.98 ± 0.30 |
| | BED | 62.73 ± 0.61 | 56.59 ± 0.08 | - | 57.20 ± 0.13 |
| | + DiD | 65.98 ± 0.40 | 60.92 ± 0.20 | - | 61.42 ± 0.21 |
| HMLP | ERM | 84.67 ± 0.64 | 55.85 ± 0.17 | 65.75 ± 0.00 | 65.05 ± 0.13 |
| | LfF | 73.33 ± 1.67 | 47.70 ± 0.58 | 54.58 ± 0.49 | 54.15 ± 0.41 |
| | + DiD | **78.67 ± 2.14** | **54.81 ± 2.26** | **63.71 ± 2.69** | **63.06 ± 2.63** |
| | BEL | 70.33 ± 2.19 | 50.96 ± 2.35 | 54.14 ± 0.25 | 54.02 ± 0.36 |
| | + DiD | 68.80 ± 0.88 | 50.20 ± 0.79 | 54.39 ± 0.18 | 54.15 ± 0.15 |
| | DisEnt | 61.67 ± 1.67 | 52.48 ± 0.56 | 54.65 ± 0.56 | 54.53 ± 0.49 |
| | + DiD | 73.67 ± 2.64 | 55.26 ± 0.93 | 62.11 ± 0.17 | 61.61 ± 0.13 |
| | BED | 75.33 ± 5.21 | 49.15 ± 1.54 | 56.86 ± 0.30 | 56.35 ± 0.35 |
| | + DiD | 78.40 ± 1.00 | 54.09 ± 1.07 | 62.05 ± 0.34 | 61.50 ± 0.38 |
| HMHP | ERM | 89.97 ± 0.34 | 29.37 ± 0.30 | - | 35.43 ± 0.24 |
| | LfF | 72.70 ± 0.81 | 35.30 ± 0.33 | - | 39.04 ± 0.33 |
| | + DiD | 82.07 ± 1.09 | 37.05 ± 0.31 | - | 41.55 ± 0.19 |
| | BEL | **82.73 ± 0.92** | 31.48 ± 0.82 | - | 36.61 ± 0.65 |
| | + DiD | 78.30 ± 0.47 | 32.90 ± 1.79 | - | 37.44 ± 1.61 |
| | DisEnt | 70.77 ± 2.27 | 36.04 ± 0.62 | - | 39.51 ± 0.36 |
| | + DiD | 76.60 ± 0.70 | **39.05 ± 0.35** | - | **42.80 ± 0.25** |
| | BED | 78.60 ± 1.56 | 34.20 ± 0.43 | - | 38.64 ± 0.38 |
| | + DiD | 78.70 ± 1.47 | 37.72 ± 0.96 | - | 41.82 ± 0.91 |

Table 7: Our approach consistently demonstrated the effectiveness on real-world datasets in both image and language modality. Group DRO is a supervised debiasing method, acting as an upper bound for worst-group accuracy.

| | Bias supervision? | MultiNLI | | CivilComments-WILDS | | CelebA | |
|---|---|---|---|---|---|---|---|
| | | Avg | Worst Acc. | Avg | Worst Acc. | Avg | Worst Acc. |
| ERM | No | 80.1 | 76.41 | 92.06 | 50.87 | 95.75 | 45.56 |
| JTT | No | 80.51 | 73.02 | 91.25 | 59.49 | 80.49 | 73.13 |
| +Ours | No | **+1.06** | **+2.71** | **+0.38** | **+6.41** | **+6.43** | **+8.50** |
| Group DRO | Yes | 82.11 | 78.67 | 83.92 | 80.20 | 91.96 | 91.49 |

Table 8: DiD still effectively boosts the performance of even very recent methods. This further demonstrated the adaptability of our approach. The experiments are conducted based on Corrupted CIFAR10.

| Algorithm | LMLP | | HMLP | | HMHP | |
|---|---|---|---|---|---|---|
| | BC acc | Avg acc | BC acc | Avg acc | BC acc | Avg acc |
| DPR | 51.54 | 54.25 | 43.67 | 44.67 | 25.92 | 31.77 |
| + DiD | **+6.97** | **+4.36** | **+5.44** | **+13.16** | **+2.61** | **+2.47** |
| DeNetDM | 60.18 | 61.98 | 49.67 | 62.11 | 24.48 | 31.25 |
| + DiD | **+1.93** | **+2.00** | **+3.66** | **+1.30** | **+3.21** | **+2.99** |

Table 9: Experiments on the multiple bias setting with LMLP and HMLP combined has demonstrated consistent effectiveness of our approach in handling multiple biases. Here HMLP BC and LMLP refer to the BC sample correctness for the HMLP-distributed bias feature and LMLP-distributed feature, respectively.

| Algorithm | LMLP BC | HMLP BC | Avg |
|---|---|---|---|
| LfF | 47.09 | 48.56 | 48.27 |
| + DiD | **+9.40** | **+10.66** | **+9.74** |
| DisEnt | 50.82 | 49.15 | 51.09 |
| + DiD | **+6.70** | **+7.01** | **+7.00** |
| BEL | 53.66 | 56.33 | 54.72 |
| + DiD | **+2.23** | **+2.89** | **+2.69** |
| BED | 56.26 | 54.66 | 56.54 |
| + DiD | **+1.02** | **+1.20** | **+0.94** |

### E.2    RESULTS ON MORE EXISTING DEBIASING METHODS

In Table 8, we show the results on more existing debiasing methods to show the generality of DiD. The results show that our approach is consistently effective on DPR and DeNetDM.

### E.3    DEBIAS ON UNBIASED DATASETS

As we do not know how biased or is the training data biased at all in real-world scenarios, it is important to evaluate the performance of debiasing methods on unbiased training data to ensure that they do not cause severe performance degradation, if not improving the performance. As shown in Table 11, existing methods perform poorly on unbiased training data, causing severe performance degradation, yet our approach greatly boosts their performances.

### E.4    EXPERIMENTS ON MULTI-BIAS SETTINGS

The existence of multiple biases is another challenge in debiasing in the real world Li et al. (2023). We further propose multiple biases with different magnitude and prevalence (HMLP + LMLP) based on Corrupted CIFAR10 benchmark, containing 10 target features and 20 spurious features. As shown in Table 9, our approach is consistently effective on multiple bias settings, debiasing multiple biases at the same time.

Interestingly, no signs of the whac-a-mole phenomenon is observed. We suspect that the phenomenon might occur only between two extremely strong biases, as assumed in the original paper.

### E.5    APPLICATION OF DiD ON THE BIAS DETECTION TASK

Some recent work (Yenamandra et al., 2023; Kim et al., 2024) has focused on the task of detecting biases rather than debiasing directly. Such methods also involve a biased auxiliary model for the

Table 10: DiD effectively improves the bias identification ability of B2T, improving both CLIP Score and Subgroup Accuracy on the ground truth bias keywords of CelebA dataset.

| | Blond: Actor | | Not Blond: Actress | |
|---|---|---|---|---|
| | CLIP Score↑ | Subgroup Acc.↓ | CLIP Score↑ | Subgroup Acc.↓ |
| B2T | 0.125 | 86.71 | 2.188 | 97.11 |
| B2T + DiD | **0.188** | **85.29** | **2.297** | **95.81** |

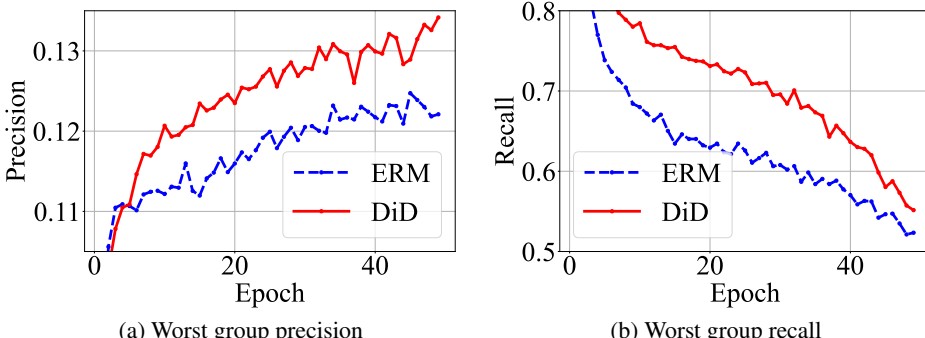

(a) Worst group precision     (b) Worst group recall

Figure 8: DiD consistently improves the worst group precision and recall in the error dataset across the epochs.

detection. To test the effectiveness of DiD on bias detection tasks, we apply DiD to the recently proposed B2T (Kim et al., 2024) method. Specifically, B2T detects bias keywords by calculating their CLIP score, whose calculation involves a biased auxiliary model to define an error dataset, similar to JTT. A keyword is identified as biased if it has a higher CLIP score and the subgroup defined by it should have lower accuracy.

Following Kim et al. (2024), we use CelebA as the dataset for bias detection, where the keyword "Actor" (a proxy for Male) is considered ground truth for class Blond, and the keyword "Actress" (a proxy for Female) is considered ground truth for the class not Blond. As we can see in Table **??**, by applying DiD to the training of the auxiliary model, we effectively improve both metrics CLIP score and subgroup accuracy, enhancing B2T's bias detection ability.

To further validate the effectiveness of DiD in improving the quality of the error dataset, we adopt the worst group precision and recall metrics proposed by Liu et al. (2021) for evaluation. Specifically, the worst group precision and recall indicate how accurately the error dataset represents the worst group samples. As shown in Figure 8, DiD improves both worst group precision and recall, demonstrating better bias identification ability.

### E.6 Results of BN samples under LMLP settings

To further examine the correctness of our analysis and the effectiveness of our design, we show the weights of BN samples under the LMLP settings. As the LMLP distribution defined in the main paper contains biased features with similar levels of bias magnitude, the choice of threshold for identifying BN samples becomes not so intuitive. Thus a threshold of 0 is selected for the categorization in the main paper, defining all samples either BA or BC samples. Consequently, we define another version of LMLP distribution named LMLP' where the magnitude of bias for each feature is low but at the same time distinguishable from each other. (Please refer to Appendix B for details) Based on LMLP' we are able to confidently define BN samples for the BN weights analysis. As shown in Figure 9, DiD consistently emphasizes BN samples in the LMLP distribution across datasets and debiasing algorithms.

Table 11: Existing debiasing methods perform poorly on unbiased training data, while DiD greatly boosts the performance.

| Algorithm | Colored MNIST | Corrupted CIFAR10 |
|---|---|---|
| ERM | 94.14 $\pm$ 0.21 | 67.91 $\pm$ 0.13 |
| LfF | 70.19 $\pm$ 1.50 | 52.04 $\pm$ 2.14 |
| + DiD | **93.18** $\pm$ 0.26 | 57.29 $\pm$ 0.22 |
| DisEnt | 75.24 $\pm$ 3.40 | 58.50 $\pm$ 0.20 |
| + DiD | 92.24 $\pm$ 0.44 | **64.58** $\pm$ 0.02 |
| BEL | 84.14 $\pm$ 0.61 | 55.64 $\pm$ 0.66 |
| + DiD | 90.02 $\pm$ 0.54 | 56.28 $\pm$ 0.45 |
| BED | 80.66 $\pm$ 0.90 | 58.57 $\pm$ 0.12 |
| + DiD | 89.10 $\pm$ 1.28 | 62.97 $\pm$ 0.16 |

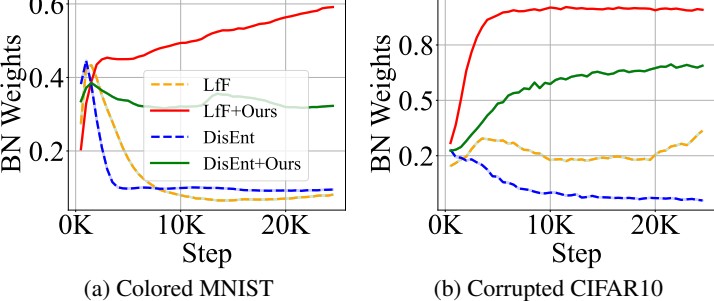

(a) Colored MNIST      (b) Corrupted CIFAR10

Figure 9: DiD consistently emphasizes BN samples in LMLP distributions across datasets and algorithms. Our approach is marked with solid lines.

Table 12: Results demonstrate that DiD is consistently effective regardless of different experimental settings of WaterBirds. The results are based on the ResNet50 architecture.

|  | Bias supervision | WaterBirds | |
| --- | --- | --- | --- |
|  |  | Avg Acc. | Worst-group Acc. |
| ERM | No | 78.82 | 31 |
| JTT | No | 90.99 | 65.26 |
| +DiD | No | **+3.45** | **+17.45** |
| Group DRO | Yes | 92.89 | 83.49 |

Table 13: Correlation of the measured dataset bias with biased model behaviour.

| Bias Magnitude - $Corr_{scp}$ | 0.1 | 0.3 | 0.5 | 0.7 | 0.9 | 0.98 |
| --- | --- | --- | --- | --- | --- | --- |
| Bias Magnitude - KLD (Ours) | 0 | 0.168 | 0.531 | 1.146 | 2.536 | 4.613 |
| Avg Acc | 93.99 | 93.50 | 92.08 | 89.53 | 77.95 | 48.13 |
| BC Acc | 93.76 | 93.11 | 91.45 | 88.55 | 75.63 | 42.42 |
| Model Bias (Avg Drop) | 0 | 0.49 | 1.91 | 4.46 | 16.04 | 45.86 |
| Model Bias (BC Drop) | 0 | 0.65 | 2.31 | 5.21 | 18.13 | 51.34 |

### E.7 ADDTIONAL RESULTS ON THE WATERBIRDS DATASET

As mentioned in Appendix D, the evaluations on the WaterBirds dataset are based on the ResNet18 architecture, which is the architecture widely adopted by many previous works (Nam et al., 2020; Lee et al., 2021; 2023). However, there are also some other works (Liu et al., 2021) that evaluate the WaterBirds dataset based on the ResNet50 architecture with better baseline performances. To demonstrate that our approach is consistently effective regardless of the experimental settings, we further test our approach with the exact same setting in Liu et al. (2021). As shown in Table 12, DiD is consistently effective regardless of different experimental settings of WaterBirds.

### E.8 CORRELATION OF THE MEASURED DATASET BIAS WITH BIASED MODEL BEHAVIOUR

We conducted additional experiments on datasets with various degrees of bias to explore how different bias magnitude measures correlate with the biased behavior of models (model bias) trained on them. Here, we measure the degree of model bias as the accuracy drop compared to the model trained on an unbiased set. As shown in Table 13, KL-divergence (KLD) measured bias magnitude strongly correlates with the model bias measured by both Average and BC sample accuracy drop, achieving a high Pearson correlation of 0.977 and 0.978, respectively. In comparison, the widely used measure $Corr_{scp} = P(y^t = a_t | y^s = a_s)$ achieves a Pearson correlation of merely 0.772 and 0.774.

### E.9 RESULTS ON REAL-WORLD TABULAR DATASETS

We further evaluate the performance of the proposed method on 3 real-world tabular datasets, including COMPAS, Adult, and German. For the COMPAS and Adult dataset, the "race" attribute is considered the biased feature. For German dataset, the attribute "sex" is considered the biased feature. For all 3 datasets in the tabular modality, we use MLP with 3 hidden layers as the backbone. As shown in the Table 14, the proposed method significantly boosts the performance of debiasing methods by a large margin on all 3 tabular datasets.

## F RELATED WORKS

**Model Bias.** The tendency of machine learning models to learn and predict according to spurious Arjovsky et al. (2020) or shortcut Geirhos et al. (2020) features instead of intrinsic features, i.e. model bias, is found in a variety of domains Heuer et al. (2016); Tang et al. (2021); Gururangan

Table 14: Performance Comparison (Avg and BC Accuracy) on COMPAS, German, and Adult Datasets

|  | COMPAS | | German | | Adult | |
| --- | --- | --- | --- | --- | --- | --- |
| **Algorithm** | **BC** | **Avg** | **BC** | **Avg** | **BC** | **Avg** |
| LfF | 40.44 | 39.35 | 46.83 | 47.06 | 51.57 | 71.94 |
| + DiD | 63.78 | **61.74** | **62.70** | 68.29 | **83.44** | 81.67 |
| DisEnt | 40.81 | 37.50 | 48.81 | 50.67 | 65.40 | 60.59 |
| + DiD | 79.56 | 49.72 | 59.52 | **71.83** | 78.68 | 80.59 |
| BEL | 48.44 | 37.71 | 43.25 | 39.00 | 46.37 | 65.35 |
| + DiD | **79.63** | 60.20 | 61.51 | 66.61 | 77.14 | **82.94** |
| BED | 48.44 | 37.22 | 44.44 | 43.83 | 48.39 | 51.67 |
| + DiD | 76.59 | 40.66 | 60.71 | 71.00 | 80.84 | 81.61 |

et al. (2018); McCoy et al. (2019); Sagawa* et al. (2020) and is of interest from both a scientific and practical perspective. For example, visual recognition models may overly rely on the background of the picture rather than the targeted foreground object during prediction. One subtopic of model bias is model fairness, which generally refers to the issue that social biases are captured by models Hort et al. (2021), where the spurious features are usually human-related and annotated, such as gender, race, and age mat; Hofmann (1994;?).

**Data Bias: spurious correlation.** Generally, spurious correlation refers to the phenomenon that two distinct concepts are statistically correlated within the training distribution, though there is no causal relationship between them, e.g. background and foreground object. The spurious correlation is a vital aspect of understanding how machine learning models learn and generalize Arjovsky et al. (2020). Specifically, studies on distribution shift Wiles et al. (2022) claim that spurious correlation is one of the major types of distribution shift in the real world, and thus an important distribution shift that a reliable model should be robust to. Furthermore, studies on fairness and bias Mehrabi et al. (2021) have demonstrated the pernicious impact of spurious correlation in classification Geirhos et al. (2019), conversation Beery et al. (2020), and image captioning Tang et al. (2021). However, despite its broad impact, spurious correlation is generally used as a vague concept in previous works and lacks a proper definition and deeper understanding of it. This is also the major motivation of this work.

**Debiasing without bias supervision.** In this work, we focus only on debiasing methods that do not require bias information, i.e. without annotation on the spurious attribute, for it is more practical. *Existing work Nam et al. (2020); Lee et al. (2021); Kim et al. (2022); Hwang et al. (2022); Lim et al. (2023); Zhao et al. (2023); Lee et al. (2023); Park et al. (2024); Han et al. (2024); Sreelatha et al. (2024) in the area generally involve a biased auxiliary model to capture biases within the training data, according to which the debiased is trained with various techniques.* Specifically, Nam et al. (2020) is the first work to propose to use GCE for bias capture, and the loss-based sample re-weighing scheme to train the debiased model. Lee et al. (2021) further proposed a feature augmentation technique to further utilize the captured bias, enhancing the BC samples. Hwang et al. (2022) proposed to augment biased data identified according to the biased auxiliary model by applying mixup Zhang et al. (2018) to contradicting pairs. Lim et al. (2023) proposed to conduct adversarial attacks on the biased auxiliary model to augment BC samples aiming to increase the diversity of BC samples. Lee et al. (2023) proposed to first filter out BC samples before training the biased auxiliary model aiming to enhance the bias capture process of the biased model. Liu et al. (2021) regards the samples misclassified by the biased auxiliary model as BC samples and emphasizes them during training of the debiased model. Park et al. (2024) proposed to provide models with explicit spatial guidance that indicates the region of intrinsic features according to a biased auxiliary model. Kim et al. (2021) create images without bias attributes using an image-to-image translation model Park et al. (2020) built upon a biased auxiliary model. A recent pair-wise debiasing method $\chi^2$ model Zhang et al. (2023a) based on biased auxiliary models encourages the debiased model to retain intra-class compactness using samples generated via feature-level interpolation between BC and BA samples.

Recently, Han et al. (2024) propose to use the disagreement leverages the disagreement probability between the target label and the prediction of a biased model to determine the weight of each training example. Sreelatha et al. (2024) propose a strategy where tehy train a deep debiased model utilizing the information acquired from both deep (perfectly biased) and shallow (weak debiased) network in the previous phase. Per-sample Gradient-based Debiasing (PGD) (Ahn & Yun, 2022) is a two-stage method that identifies bias-conflicting (BC) samples by their high gradient norms from an auxiliary biased model, then resamples the data to focus the final model on these "hard" samples. A more recent and conceptually related approach is DiffuBias (Ko et al., 2024), which also begins by training an auxiliary biased model to capture biases via top-K loss. DiffuBias employs an augmentation pipeline: it captions the identified hard samples using an LLM and then uses a latent diffusion model to generate new, synthetic bias-conflict images from these text prompts. *Despite different technique routes, it's implicitly or explicitly assumed by these works that the biases can be well captured by the biased models, which serves as a foundation for subsequent debiasing. However, in this work, we show that such assumptions might be challenged under real-world scenarios.*

## G  LIMITATIONS AND FUTURE WORK

We uncover the insufficiency of existing debiasing benchmarks theoretically and empirically, highlighting the importance and novel challenges of debiasing in the real world, i.e. Sparse bias capturing. We further proposed a simple yet effective method to address the challenge. However, there are still a few limitations of this work:

- While we have proposed fine-grained empirical and theoretical analysis on real-world biases with important characteristics found, due to the complexity of the data bias problem, there might be other important characteristics of real-world biases that we are unaware of. We believe this is another important direction for future research, which serves as the foundation for developing debiasing methods that are applicable in the real world.

- While DiD has been shown to be simple and effective, it remains a preliminary solution to tackle the Sparse bias capturing challenge in real-world debiasing, and there is still much room for improvement. We believe there will be more sophisticated and potentially better-performing approaches to tackle the challenge in the future.

We see potential within those limitations and leave them for future research.

## H  BOARDER IMPACT

From a technical standpoint, our research provides a fine-grained framework for analyzing data biases, a systematic evaluation framework for real-world debiasing, and a simple yet effective solution to the challenges in real-world debiasing. The bias analysis framework serves as the basis for deepen our understanding of dataset biases. The evaluation framework paves the path towards developing debiasing methods applicable in real-world scenarios. The proposed approach DiD, is highly effective and adaptive to the various debiasing methods. Thus, we believe DiD have high potential to be adopted in debiasing methods in the future.

By advancing the understanding of dataset biases and improving the performance of debiasing methods in real-world scenarios, our research contributes to the development of more robust and generalizable AI models. This is particularly relevant in an era where AI systems are increasingly deployed in dynamic and diverse environments, necessitating models that can adapt and maintain high performance across different contexts and populations.

Table 15: Our approach still effectively boosts the performance of even very recent methods. This further demonstrated the adaptability of our approach. The experiments are conducted based on Corrupted CIFAR10.

| Algorithm | LMLP | | HMLP | | HMHP | |
|---|---|---|---|---|---|---|
| | BC acc | Avg acc | BC acc | Avg acc | BC acc | Avg acc |
| DPR | 51.54 | 54.25 | 43.67 | 44.67 | 25.92 | 31.77 |
| + DiD | **+6.97** | **+4.36** | **+5.44** | **+13.16** | **+2.61** | **+2.47** |

