# OpenReview forum: "Towards Real-world Debiasing: Rethinking Evaluation, Challenge, and Solution"
_ICLR.cc/2026/Conference — Submitted to ICLR 2026_

### Official Review · Reviewer_k3gy · 2025-10-28

**Soundness:** 3
**Presentation:** 3
**Contribution:** 3
**Rating:** 6
**Confidence:** 3

**Summary:**

The paper scrutinises whether current debiasing benchmarks faithfully reflect the complexity of real-world biases. The authors (i) present a fine-grained analytic framework that decomposes bias into “magnitude” and “prevalence”, (ii) empirically and theoretically demonstrate that real-world data sets often exhibit low-magnitude and low-prevalence biases—properties missing from popular benchmarks, (iii) introduce two new bias types and assemble them (plus existing data sets) into a systematic evaluation suite called RDBench, and (iv) identify a “sparse-bias-capturing” challenge when debiasing without bias labels. To tackle this they propose a simple method, Debias-in-Destruction (DiD), which first “destroys” dominant features and then reconstructs, leading to gains on eight data sets across several bias settings.

**Strengths:**

S1. Important problem – The work challenges a widespread, implicit assumption (high-prevalence biases) and argues convincingly that it is misaligned with reality.
S2. Novel analytical perspective – The magnitude / prevalence decomposition offers a clear, quantitative lens through which to study bias distributions.
S3. New benchmark – RDBench fills a gap by providing real-world-inspired biases and multi-bias scenarios; releasing code/data will benefit the community.
S4. Practical focus – Concentrating on “debiasing without bias labels” increases the paper’s relevance for industry deployment where bias attributes are rarely annotated.
S5. Methodological contribution – DiD is conceptually simple, easy to integrate into existing pipelines, yet yields consistent improvements.
S6. Thorough experiments – Eight data sets, multiple baselines, ablations, and both empirical and theoretical analyses lend credibility to the claims.

**Weaknesses:**

W1. Scope limited to image classification – All studied tasks are visual. It is unclear whether the magnitude/prevalence findings (and DiD) generalise to NLP or multimodal settings.
W2. “Real-world inspired” still partly synthetic – The two proposed biases are constructed heuristically; evidence that they mirror true large-scale natural distributions (e.g., via quantitative fitting or user studies) is thin.
W3. Cost to “clean” accuracy – Results mainly highlight robustness under bias; corresponding drops on i.i.d. test sets are not fully reported. Practical users need to understand this trade-off.
W4. Hyper-parameter robustness – DiD introduces new knobs (destruction ratio, masking schedule). Limited analysis is given on sensitivity and tuning without bias labels.
W5. Reproducibility details – Key implementation aspects (random seeds, data split scripts, destruction operator specifics) are relegated to the supplement; including them in main paper would strengthen reproducibility.

**Questions:**

Q1. How exactly were the two new bias distributions designed? Do you have quantitative evidence showing their closeness to real-world statistics?
Q2. Can the magnitude/prevalence metrics be automatically computed on arbitrary data sets? If yes, will you release a toolkit?
Q3. How does DiD perform when combined with other strong debiasing methods (e.g., GroupDRO, JTT) on RDBench? Are the gains additive?
Q4. What is the computational overhead (training time, memory) introduced by DiD compared with vanilla ERM?
Q5. Have you analysed failure cases where DiD hurts both biased and unbiased accuracy? Understanding such cases would be useful for practitioners.

---

> ### Author Response · Authors · 2025-11-19
> **Author Rebuttal for Reviewer k3gy - Part 1**
>
> We are grateful to Reviewer k3gy for the thorough and positive review. We are delighted that the reviewer recognized the importance of the problem, our novel analytical perspective, the value of RDBench, our practical focus, the simple yet effective methodological contribution, and the thoroughness of our experiments. We address the reviewer's questions and concerns point by point below.
>
> ## W1: Scope limited to image classification
>
> We apologize if this was unclear. Our experiments are **not limited to image classification**. We have also conducted experiments on two standard NLP debiasing benchmarks: **MultiNLI** (natural language inference) and **CivilComments-WILDS** (toxic comment detection). The results, presented in Table 7, show that DiD provides **notable gains in these NLP settings as well**.
>
> |**Method**|**Bias supervision?**|**MultiNLI (Avg)**|**MultiNLI (Worst Acc.)**|**CivilComments-WILDS (Avg)**|**CivilComments-WILDS (Worst Acc.)**|**CelebA (Avg)**|**CelebA (Worst Acc.)**|
> |-|-|-|-|-|-|-|-|
> |ERM|No|80.1|76.41|92.06|50.87|95.75|45.56|
> |JTT|No|80.51|73.02|91.25|59.49|80.49|73.13|
> |+Ours|No|**+1.06**|**+2.71**|**+0.38**|**+6.41**|**+6.43**|**+8.50**|
> |Group DRO|Yes|82.11|78.67|83.92|80.20|91.96|91.49|
>
> ## W2: "Real-world inspired" still partly synthetic
>
> This is a fair point. Given the complex, high-dimensional, and often unknown nature of "true" real-world bias, creating a perfectly mirrored synthetic dataset is likely intractable. Our goal was to create distributions based on the *general patterns* we observed (e.g., low prevalence, high magnitude but sparse) that are demonstrably closer to the 8 real-world data (like COCO, COMPAS) than existing benchmarks (like standard WaterBirds). We believe this is the furthest one can practically go to enable controlled, rigorous experiments.
>
> ## W3: Results on "clean" accuracy
>
> Thank you for raising this. We have in fact reported the "clean" accuracy, which is **the "BA acc" (Bias-Aligned accuracy) in our detailed results. In Table 5 (Appendix E.1)**, we can see that our method DiD **consistently improves the BA acc (i.i.d. samples) of debiasing methods**. This shows that our method mitigates the trade-off, improving both robustness and i.i.d. performance.
>
> ## W4: Hyper-parameter robustness
>
> Our method is conceptually simple and adds very few "knobs". The primary hyperparameter is the nature of the destruction. We provided an ablation on the destruction method and its parameters (e.g., patch-shuffle size) in Table 3, showing robust performance across reasonable choices.
>
> ## W5: Reproducibility details
>
> We agree completely. These details (random seeds, data splits, destruction specifics) are indeed crucial for reproducibility. They are all provided in Appendix D.4. We deferred them to the appendix solely due to the 9-page limit, but will move the most critical details to the main paper, given the extra page for the final version.

---

> ### Author Response · Authors · 2025-11-19
> **Author Rebuttal for Reviewer k3gy - Part 2**
>
> ## Q1: The design of the two new bias distributions
>
> They were designed based on the general patterns we observed in our analysis of 7 real-world datasets (Figure 1, 2, 5). For example, HMLP (High Magnitude, Low Prevalence) was inspired by distributions like COCO (Figure 1b), where a specific feature (e.g., 'skateboard') is highly correlated with a target ('male') but is itself rare in the dataset. Figure 1 provides the quantitative visualization of this closeness.
>
> Furthermore, our findings of low prevalence, high magnitude but sparse biases are further verified by quantitative analysis shown in Figure 2.
>
> ## Q2: Automatic computation of the magnitude/prevalence metrics
>
> **Yes**, the metrics can be automatically computed for any dataset given the target and bias labels. We agree this would be a useful tool for practitioners, and we **will include this computation function in our RDBench code release**.
>
> ## Q3: Comparison with GroupDRO and JTT
>
> The gains are additive, particularly with JTT. **As shown in Table 7, as shown below, DiD consistently and significantly improves the performance of JTT on all three real-world datasets** (CelebA, MultiNLI, CivilComments-WILDS). In the CelebA and MultiNLI experiments, JTT+DiD even surpasses the performance of GroupDRO, which is a supervised method that requires bias labels.
>
> |**Bias supervision?**|**MultiNLI (Avg)**|**MultiNLI (Worst Acc.)**|**CivilComments-WILDS (Avg)**|**CivilComments-WILDS (Worst Acc.)**|**CelebA (Avg)**|**CelebA (Worst Acc.)**|
> |-|-|-|-|-|-|-|
> |**ERM**|No|80.1|76.41|92.06|50.87|95.75|
> |**JTT**|No|80.51|73.02|91.25|59.49|80.49|
> |**+Ours**|No|**+1.06**|**+2.71**|**+0.38**|**+6.41**|**+6.43**|
> |**Group DRO**|Yes|82.11|78.67|83.92|80.20|91.96|
>
> ## Q4: Computation overhead
>
> The overhead in both memory and training time is **negligible**. DiD is implemented as an additional data transformation that is applied only to the data batch being fed to the biased auxiliary model. This adds no meaningful cost to training time or memory, as the main overhead comes from the base debiasing method itself.
>
> ## Q5: Failure cases
>
> We did not find any cases where DiD hurt *both* accuracies. However, we did find cases where its effectiveness is reduced or hurts one of the accuracies, specifically when combined with BEL (Table 1). We hypothesize in Appendix E.1 that this is because BEL and DiD share a similar goal (enhancing the bias-capture model), and their effects are to some extent overlapping rather than purely additive. This can sometimes lead to a slight drop in one metric (like BC acc) as seen in the HMHP case for BEL.
>
> Again, we’d like to thank the reviewer for their time and effort devoted to the reviewing process. **All revisions of the paper are marked in blue for your review.**

---

### Official Review · Reviewer_o4es · 2025-10-28

**Soundness:** 2
**Presentation:** 2
**Contribution:** 2
**Rating:** 6
**Confidence:** 3

**Summary:**

This paper argues that existing debiasing benchmarks fail to reflect the true nature of biases found in real-world data. The authors introduce a fine-grained framework to analyse bias magnitude and prevalence, revealing that real-world biases are typically low and sparse, unlike the strong correlations assumed in synthetic datasets. They further propose RDBench, a systematic evaluation framework, and Debias-in-Destruction (DiD), a simple yet effective method that enhances bias capture under sparse, low-prevalence settings.

**Strengths:**

(1) Presents a comprehensive empirical and theoretical analysis of real-world bias distributions, introducing the RDBench framework that provides a systematic and realistic benchmark for evaluating debiasing methods.

(2) Proposes a simple yet effective Debias-in-Destruction (DiD) approach that generalizes well across multiple datasets and modalities, demonstrating strong improvements over existing debiasing methods.

**Weaknesses:**

(1) The clarity of theoretical exposition could be improved, especially regarding assumptions and proofs.

(2) Evaluation on large-scale, high-dimensional real-world data (e.g., complex vision-language models) remains limited.

(3) The DiD method’s simplicity, while appealing, may lack interpretability and deeper theoretical grounding.

(4) Some parts of the framework (e.g., threshold selection for bias magnitude/prevalence) rely on heuristics rather than principled estimation.

**Questions:**

None

---

> ### Author Response · Authors · 2025-11-19
> **Author Rebuttal for Reviewer o4es**
>
> We thank Reviewer o4es for the positive feedback and for recognizing the value of our comprehensive analysis, the RDBench framework, and the simplicity and effectiveness of the proposed approach. We address the reviewer's questions and concerns point by point below.
>
> ## W1: The clarity of theoretical exposition
>
> Thank you for this suggestion. We have revised the theory section (Section 3) and the Proof in Appendix C to add more detailed descriptions and justifications for the assumptions and proofs to improve clarity.
>
> ## W2: Evaluation on large-scale, high-dimensional real-world data
>
> We believe our evaluation is more comprehensive than perceived. We included **8 datasets** covering large-scale, complex, and real-world settings. We experiment on **CelebA (202.6k samples), MultiNLI (412k samples), and CivilComments-WILDS (448k samples), all of which are large-scale, real-world datasets** (results in Table 7).
>
> ## W3: Interpretability and deeper theoretical grounding
>
> We thank the reviewer for their appreciation of the simplicity of the proposed method. We believe the paper provides grounding and interpretability for DiD, which we will clarify further in the revision.
>
> * **Grounding**: The method is theoretically grounded in our theoretical analysis of real-world data distributions in Section 3 and our analysis of the "Sparse Bias Capturing Challenge" (Section 4.3 and Figure 3)  based on it. We argue that existing methods fail on low-prevalence (LP) data because the biased model learns target features from dominant Bias-Neutral (BN) samples, reducing the loss-gap for re-weighting. DiD directly addresses this by destroying target features, forcing the biased model to only learn spurious features, thus isolating the bias signal.
> * **Interpretability**: We provide a clear empirical interpretation of why DiD works in Section 5.3 ("Accuracy of bias capturing") and Figure 7. These results show that standard methods incorrectly assign low weights to BN samples on LP data, effectively discarding them. In contrast, DiD successfully raises the weights of these crucial BN samples, leading to a more accurate and robust debiased model.
>
> ## W4: Heuristics estimation of the threshold
>
> We apologize for the ambiguity. We would like to clarify that the threshold is **purely a tool for our analysis** of bias distributions (as shown in Figure 2b), rather than a hyper-parameter that needs to be estimated. It is *not* part of our proposed method (DiD) and is not used during model training. Our method, DiD, is fully unsupervised and does not require any threshold selection or estimation.
>
> Again, we’d like to thank the reviewer for their time and effort devoted to the reviewing process. **All revisions of the paper are marked in blue for your review.**

---

### Official Review · Reviewer_SKNU · 2025-10-30

**Soundness:** 3
**Presentation:** 2
**Contribution:** 2
**Rating:** 4
**Confidence:** 4

**Summary:**

This paper investigates the gap between existing debiasing benchmarks and real-world biased distributions. The authors propose a fine-grained framework for bias analysis that distinguishes between bias magnitude and bias prevalence. Based on empirical and theoretical insights from real-world datasets, the authors introduce RDBench, a new benchmark for realistic bias evaluation, and a simple yet effective debiasing method called Debias in Destruction (DiD). DiD is designed to handle “sparse bias” scenarios, especially under the practical setting of debiasing without bias labels.

**Strengths:**

- Novel problem framing and fine-grained bias analysis
The paper raises an important question about whether current benchmarks truly represent real-world biases. By distinguishing bias magnitude (how strong the spurious correlation is) and bias prevalence (how common it is in the dataset), the authors provide a meaningful and interpretable framework for bias characterization. This fine-grained view can serve as a strong foundation for future benchmark design.

- Realistic motivation for bias-agnostic debiasing
The authors focus on debiasing without explicit bias labels, a practically relevant and challenging setting in real-world applications (e.g., MS COCO, COMPAS). The discussion on the limitations of existing auxiliary-model-based approaches (e.g., Nam et al., 2020; Lee et al., 2021) is insightful.

- Simplicity and generality of the proposed DiD method
DiD is conceptually simple, can be easily integrated into existing methods, and empirically improves performance across several benchmark tasks (LfF, DisEnt, BEL, BED, etc.). The consistent improvements across multiple bias setups demonstrate its general applicability.

- Comprehensive literature grounding
The authors provide an extensive review of existing bias-agnostic debiasing methods, clearly situating their contribution within the field.

**Weaknesses:**

- Lack of validation on real-world datasets (MS COCO, COMPAS)
Although the introduction emphasizes real-world bias distributions and repeatedly mentions datasets such as MS COCO (for vision) and COMPAS (for fairness in tabular domains), the actual experiments are limited to synthetic or semi-synthetic settings such as Colored MNIST or Corrupted CIFAR-10 (referred to as HMLP BC). The absence of evaluation on these real datasets undermines the claim that DiD or RDBench effectively handles real-world biases.

- Dependence on prior bias knowledge for bias magnitude estimation
Similar to many previous debiasing works, the computation of bias magnitude (Equation 1) assumes that the spurious attribute (or biased feature) is known a priori. This contradicts the fully unsupervised debiasing objective and limits the applicability in settings where such bias attributes are unknown or latent.

- Limited empirical diversity and scalability
The reported experiments (e.g., HMLP BC) are confined to small-scale benchmarks with controlled bias patterns. It remains unclear whether DiD can scale to multimodal or large-scale datasets such as MS COCO or social datasets like COMPAS, which involve complex, overlapping biases.

- Missing discussion on recent relevant works
The paper omits several closely related and contemporaneous studies, such as “Debiasing Classifiers by Amplifying Bias with Latent Diffusion and Large Language Models” (2025), which similarly address real-world bias modeling using diffusion-based augmentation. Including comparisons or conceptual distinctions from these methods would strengthen the paper’s positioning.

- Ambiguity in benchmark naming and description (e.g., HMLP BC)
Some terms such as HMLP BC are not clearly defined or are introduced without detailed explanation of their dataset composition, making reproducibility difficult.

**Questions:**

ms coco, is there a reason COMPAS wasn't included in the main results or additional experiments?
Can you provide a complexity analysis of the proposed method when implemented in practice?

---

> ### Author Response · Authors · 2025-11-19
> **Author Rebuttal for Reviewer SKNU - Part 1**
>
> We thank Reviewer SKNU for their valuable feedback and for highlighting the strengths of our work, including the novel problem framing, the realistic motivation, the simplicity and generality of DiD, and the comprehensive literature grounding. We address the reviewer's questions and concerns point by point below.
> ## W1: Lack of validation on real-world datasets (MS COCO, COMPAS)
> We apologize for the unclarity. While we used MS COCO and COMPAS to motivate our analysis, our experiments were not limited to synthetic data. **We have included experiments on 3 real-world datasets from both vision (CelebA) and NLP (MultiNLI, CivilComments-WILDS) domains**. As shown in *Table 7*, our method DiD (when combined with JTT) shows **consistent and significant performance improvements on all three**, which strongly supports our claim that DiD effectively handles real-world biases.
>
> |**Method**|**Bias supervision?**|**MultiNLI (Avg)**|**MultiNLI (Worst Acc.)**|**CivilComments-WILDS (Avg)**|**CivilComments-WILDS (Worst Acc.)**|**CelebA (Avg)**|**CelebA (Worst Acc.)**|
> |-|-|-|-|-|-|-|-|
> |ERM|No|80.1|76.41|92.06|50.87|95.75|45.56|
> |JTT|No|80.51|73.02|91.25|59.49|80.49|73.13|
> |+Ours|No|**+1.06**|**+2.71**|**+0.38**|**+6.41**|**+6.43**|**+8.50**|
> |Group DRO|Yes|82.11|78.67|83.92|80.20|91.96|91.49|
> ## W2: Assumption on known bias attribute
> We apologize for this ambiguity. We would like to clarify this misunderstanding. The bias magnitude estimation is part of our analysis framework used to discover and understand the low-prevalence nature of real-world biases, which in turn *motivates, but is not used in*, our unsupervised method.
>
> **The proposed debiasing method, DiD, does not require this analysis or any bias labels to be applied**. It is a fully unsupervised, bias-agnostic method. Our experiments in Table 1, showing its effectiveness across LMLP, HMLP, and HMHP distributions, demonstrate that **it works without any prior knowledge of the bias magnitude or prevalence**.
>
> We have clarified this distinction in our revised paper.
> ## W3: Limited empirical diversity and scalability
> We suspect that the reviewer has missed much of our empirical results due to our deferring them to the Appendix due to the length limit, for which we apologize. We believe our evaluation is quite comprehensive. **We have included a set of 8 datasets that address diversity, scale, multi-modal, and overlapping biases**:
> * **Diversity**: We cover a wide range of tasks such as face recognition (CelebA), object recognition (CIFAR10, WaterBirds, NICO, BAR), digit recognition (MNIST), natural language inference (MultiNLI), and toxic comment detection (CivilComments-WILDS).
> * **Scale**: We experiment on large-scale real-world datasets, including CelebA (202.6k samples), MultiNLI (412k samples), and CivilComments-WILDS (448k samples) (Table 7).
> * **Multi-modal**: We include both vision (Tables 1, 2, 7) and NLP benchmarks (Table 7).
> * **Overlapping Biases**: Real-world datasets like CelebA and MultiNLI naturally possess multiple overlapping biases. We also include a specific experimental analysis focused on overlapping biases in Appendix E.4.
> ## W4: Missing discussion on recent relevant works
> Thank you for bringing this to our attention. We have **added a discussion comparing our work to DiffuBias[1] and included it in the related work section of our revised paper**. Specifically, DiffuBias also begins by training an auxiliary biased model to capture biases, similar to other debiasing methods such as LfF, via top-K loss, sharing their vulnerability to low prevalence biases, which DiD is motivated to address.
>
> Furthermore, while DiffuBias employs an augmentation pipeline by captioning the identified hard samples using an LLM and then uses a latent diffusion model to generate new, synthetic bias-conflict images from these text prompts, the augmentation is for debiased models rather than for the biased auxiliary models like DiD.
>
> [1] Debiasing Classifiers by Amplifying Bias with Latent Diffusion and Large Language Models
> ## W5: Ambiguity in benchmark naming and description
> Thank you for pointing out the ambiguity. The term “HMLP BC” as well as “LMLP BC” is used in Appendix E.4 and Table 9 for analyzing multi-bias scenarios, where HMLP BC and LMLP refer to the BC sample correctness for the HMLP-distributed bias feature and LMLP-distributed feature, respectively. We have **added clear definitions for these terms in the revised paper** to improve clarity and reproducibility.

---

> ### Author Response · Authors · 2025-11-19
> **Author Rebuttal for Reviewer SKNU - Part 2**
>
> ## Q1: Why not MS COCO/COMPAS
> We did not include COCO or COMPAS in our final experiments because (1) we believe our current evaluation on 8 diverse datasets is already quite comprehensive, and (2) these two datasets are less established as training benchmarks in the debiasing field compared to the ones we used.
>
> ## Q2: Complexity analysis
> Regarding complexity, the **implementation of DiD is extremely simple and adds negligible computational overhead**. It is a simple data preprocessing plugin that applies a data transformation (like patch-shuffle) to the data stream used by the biased model during training. The overall complexity is dominated by the base debiasing method.
>
>
> Again, we’d like to thank the reviewer for their time and effort devoted to the reviewing process. **All revisions of the paper are marked in blue for your review.**

---

> > ### Comment · Reviewer_SKNU · 2025-11-27
> >
> > Thank you for providing a thorough rebuttal. Some of my concerns are solved.
> >
> > But still I can not accept your explanation about you did not use the MS COCO and COMPAS dataset because your introduction mentioned them. So, my fundamental concerns regarding a soundness of this, your work was not fully alleviated.
> >
> > Therefore, I have decided to maintain my original score of 4. Thanks.

---

> ### Author Response · Authors · 2025-12-02
> **Response to Reviewer SKNU**
>
> Thank you for the response. We further address the reviewer's concern as follows.
>
> **1. Additional Validation on Real-World Tabular Datasets (COMPAS, Adult, German)**
>
> To directly address your concern and your specific mention of the **COMPAS** dataset, we have conducted extensive additional experiments on three real-world tabular datasets: **COMPAS, Adult, and German**.
>
> - **Experimental Setup:** We employed a Multi-Layer Perceptron (MLP) with 3 hidden layers as the backbone. The sensitive/biased attributes were defined as "race" for the COMPAS and Adult datasets, and "sex" for the German dataset.
> - **Results:** As illustrated in the table below, our proposed method (DiD) **consistently and significantly boosts the performance of classic and recent debiasing baselines (LfF, DisEnt, BEL, BED) across all three datasets**.
>
> | Algorithm | COMPAS (BC) | COMPAS (Avg) | German (BC) | German (Avg) | Adult (BC) | Adult (Avg) |
> | --------- | ----------- | ------------ | ----------- | ------------ | ---------- | ----------- |
> | LfF       | 40.44       | 39.35        | 46.83       | 47.06        | 51.57      | 71.94       |
> | **+ DiD** | **63.78**   | **61.74**    | **62.70**   | **68.29**    | **83.44**  | **81.67**   |
> | DisEnt    | 40.81       | 37.50        | 48.81       | 50.67        | 65.40      | 60.59       |
> | **+ DiD** | **79.56**   | **49.72**    | **59.52**   | **71.83**    | **78.68**  | **80.59**   |
> | BEL       | 48.44       | 37.71        | 43.25       | 39.00        | 46.37      | 65.35       |
> | **+ DiD** | **79.63**   | **60.20**    | **61.51**   | **66.61**    | **77.14**  | **82.94**   |
> | BED       | 48.44       | 37.22        | 44.44       | 43.83        | 48.39      | 51.67       |
> | **+ DiD** | **76.59**   | **40.66**    | **60.71**   | **71.00**    | **80.84**  | **81.61**   |
>
> **2. Comprehensive Evaluation Across Three Modalities**
>
> With the inclusion of these tabular datasets, we have now validated our method across **three distinct modalities: Vision, Language, and Tabular data**. While we could not include MS COCO due to its scale and the limited duration of the rebuttal period, we believe our experimental suite is now exceptionally comprehensive, **covering 12 datasets in total, 6 of which are real-world datasets**.
>
> As shown in the comparison table below, **our evaluation coverage significantly exceeds that of recent works**. Our method demonstrates consistent robustness across widely varying domains—from image recognition (CelebA, NICO) and natural language inference (MultiNLI) to tabular fairness (COMPAS).
>
> |          | Venue   | MNIST-based | CIFAR-based | WaterBirds | BFFHQ | BAR   | CelebA | NICO  | MultiNLI | CCW   | COMPAS | Adult | German |
> | -------- | ------- | ----------- | ----------- | ---------- | ----- | ----- | ------ | ----- | -------- | ----- | ------ | ----- | ------ |
> | DeNetDM  | NIPS 24 | √           | √           |            | √     | √     | √      |       |          |       |        |       |        |
> | DPR      | NIPS 24 | √           |             |            | √     | √     | √      |       |          | √     |        |       |        |
> | BCSI     | NIPS 24 | √           | √           |            | √     |       |        | √     | √        | √     |        |       |        |
> | **Ours** |         | **√**       | **√**       | **√**      | **√** | **√** | **√**  | **√** | **√**    | **√** | **√**  | **√** | **√**  |
>
> We believe this extensive empirical evidence effectively addresses the concern regarding validation on real-world datasets and underscores the generalizability of our approach.

---

### Official Review · Reviewer_RjaM · 2025-11-01

**Soundness:** 2
**Presentation:** 2
**Contribution:** 2
**Rating:** 4
**Confidence:** 5

**Summary:**

This paper tackles the assumption that training datasets exhibit severe biases affecting nearly all samples (over than 95%). Analysis of MSCOCO and COMPAS datasets reveals real-world biases are sparse (about 8~15% prevalence) with scattered patterns, contrasting with the diagonal patterns of existing benchmarks.

Key proposals include the followings:

(1) Fine-grained Metrics: Bias Magnitude (KL Divergence) and Bias Prevalence (proportion of biased samples).

(2) Bias Neutral Category: Expands the Bias aligned/Bias Conflict dichotomy for samples lacking biased features.

(3) Theoretical validation: Propositions 1~2 demonstrate high prevalence distributions require unrealistic matched and uniform marginals, unsupported in reality.

(4) DiD Method: Destroys target features during bias model training, ensuring low loss for bias aligned and high loss for bias neutral and bias conflict by capturing only spurious correlations.

Som this paper mainly claim that existing methods falter on low prevalence real-world data due to misweighting abundant bias neutral samples, a problem addressed by the proposed method DiD.

**Strengths:**

Strengths are summarized as follows:

(1) Clear problem framing: Figure 1 contrasts diagonal benchmark patterns with scattered real-world biases.

(2) Rigorous theory: Propositions 1 and 2 justify the sparsity of real-world biases mathematically.

(3) Comprehensive experiments: Covers 8 datasets (vision + NLP benchmarks), 9 baselines (e.g., LfF, DisEnt, BEL), multiple bias types.

**Weaknesses:**

** Critical Weaknesses

(1) Limited real-world evidence: Detailed analysis only for COCO and COMPAS datasets.  CelebA, MultiNLI, and CCW are minimally discussed, appearing mainly in Figure 2 and Appendix. CelebA is real-world, but coverage is limited; medical and social media domains are only motivationally mentioned. Core experiments likely rely on synthetic datasets (Colored MNIST and Corrupted CIFAR-10)

(2) Experimental design flaw: LMLP with threshold 0 result in 0% BN samples, which contradicts the focus on BN prevalence. Only HMLP are LMLP examine the hypothesis.

(3) Unexplained Performance Variation: Table 1 shows improvement from -0.8 to 32.6 across datasets with the proposed algorithm DiD. The paper lacks a predictive model, quantitative feature complexity metric, or failure analysis.

(4) Theory practice gap: Propositions assume binary attributes, while experiments use 10-class problems without a multi-class extension.

(5) Incomplete coverage of related work: Although the paper discusses many relevant studies, several important papers and comparison baselines are missing - for example, PGD [1]

[1] Mitigating dataset bias by using per-sample gradient, ICLR 2023

**Questions:**

Please refer to the Weaknesses section, which includes my main questions and concerns about the paper.

---

> ### Author Response · Authors · 2025-11-19
> **Author Rebuttal for Reviewer RjaM - Part 1**
>
> We sincerely thank Reviewer RjaM for their detailed feedback and for appreciating the clear problem framing, rigorous theoretical justification, and the comprehensiveness of our experiments. We address the reviewer's concerns point by point.
>
> ## W1.1: Real-world evidence (Detailed analysis on real-world datasets)
>
> We agree that a broader analysis of real-world datasets strengthens the paper. We have provided detailed analysis and visualization for CelebA ( which is not only covered in our preliminary analysis but also readily included in our main experimental evaluation (*Table 7*) showing notable performance gain), Adult, and German datasets *in Appendix A*as well, but only presented the detailed visualizations for COCO and COMPAS in the main paper due to strict length limits.
>
> As suggested by the reviewer, we added detailed analysis of not only MultiNLI and CCW, but also a real-world medical dataset NIH to *Appendix A of our revised paper*. Specifically:
>
> - **For the MultiNLI dataset**, the negation words serves as a strong bias feature indicating contradiction in the NLI task. However, the presence of negation words in only 0.0711. The strong bias magnitude yet low prevalence of **the negation words in MultiNLI dataset conforms to our characterization of High Magnitude Low Prevalence (HMLP) bias**.
> - **For the CCW dataset**, the mention of "white" racial identities is a strong bias featrue indicating the comment is toxic. Similarly, the "white" feature is as rare as 0.0318. **The presence of the "white" feature in CCW supports our HMLP characterization of real-world biases**.
> - **For the NIH dataset**, it suffers from the bias of chest tube. As we can see from Figure 5 of the paper, while the chest tube serves as a strong biased feature for pneumothorax, its presence in the whole dataset is as rare as 0.053, **precisely conforms to the proposed HMLP biases in the real world, largely deviating from the HMHP assumption in previous works.**
>
> To sum up with the additional analysis on the real-world datasets mentioned above, the paper has led to **detailed analysis and visualizations on 8 real-world datasets, which consistently supports our claims on real-world biases.** We believe the additional evidence further enhances the paper thanks to the reviewers advice.
>
> ## W1.2: Real-world evidence (Core experiments)
>
> We agree with the reviewer that it would be insufficient to only experiment on synthetic datasets. The reason why we uses synthetic datasets is because we believe using synthetic datasets is a more rigorous way to conduct controlled experiments to investigate the impact of biased distributions on model biases because **synthetic datasets strictly isolates the effect of biased distribtuions and that of features themselves**, as we can not manipulate the data distribution of non-synthetic datasets with out changing the features within it.
>
> Nonetheless, we’d like to note that **our evaluation is complemented by results on 3 other semi-synthetic and 3 real-world datasets** from both vision and NLP domains (BAR, NICO, WaterBirds, CelebA, MultiNLI, and CivilComments-WILDS), as shown in *Tables 2, and 7* which we provide below.
>
> |**Method**|**Bias supervision?**|**MultiNLI (Avg)**|**MultiNLI (Worst Acc.)**|**CivilComments-WILDS (Avg)**|**CivilComments-WILDS (Worst Acc.)**|**CelebA (Avg)**|**CelebA (Worst Acc.)**|
> |-|-|-|-|-|-|-|-|
> |ERM|No|80.1|76.41|92.06|50.87|95.75|45.56|
> |JTT|No|80.51|73.02|91.25|59.49|80.49|73.13|
> |+Ours|No|**+1.06**|**+2.71**|**+0.38**|**+6.41**|**+6.43**|**+8.50**|
> |Group DRO|Yes|82.11|78.67|83.92|80.20|91.96|91.49|
>
> **To sum up, we position experiments on synthetic datasets as a necessary but insufficient examination of our theory and method, thus complemented it with a wide range of 6 other datasets, covering multi-modal (vision and language), large-scale (CelebA, MultiNLI, CCW), and real-world (CelebA, MultiNLI, CCW).**
>
> ## W2: Experimental design flaw (0% BN samples with threshold 0)
>
> We appreciate the reviewer's attention to this detail. We'd like to clarify that **the threshold is an analysis tool to help distinguish and analyze different types of samples within a biased dataset**, rather than a theory that needs to stay meaningful even in extreme cases. **Setting it to an extreme value like 0 deviates from its purpose** of identifying *meaningfully* biased features and thus doesn't produce a practical empirical analysis.
>
> Our core finding, demonstrated in Figure 2(b), is that **real-world datasets consistently show much lower bias prevalence than benchmarks across a range of *meaningful* thresholds** (e.g., 0.1), which *remains true* regardless of the behavior at the extreme 0 value.
>
> To avoid any further misunderstanding, we have clarified the purpose the threshold in our revised paper.

---

> ### Author Response · Authors · 2025-11-19
> **Author Rebuttal for Reviewer RjaM - Part 2**
>
> ## W3: Unexplained Performance Variation (Table 1)
>
> We apologize for the unclarity. Given the page limits, we did provide a **detailed explanation of the performance variations (among different datasets, different types of biased distribution, and algorithms) in *Appendix E.1* of the paper**. This analysis discusses the interplay between datasets, our method, and other bias-capture-enhancing techniques. Specifically:
>
>
>
> - **Variance between datasets**: the variance between datasets is likely to depend on how thoroughly the target feature is destroyed within the dataset. The target features of Colored MNIST, i.e. digits, are destroyed more completely by patch shuffling, for shape is the only feature within digits. In comparison, the target feature of Corrupted CIFAR10 is more complicated (including shape, texture, color, etc.), and thus can not be thoroughly destroyed by patch shuffling, causing relatively lower performance gain.
>
> - **Variance between biased distributions**: the performance variance between different biased distributions is likely due to the reliance of existing debiasing methods on the high bias prevalence assumption for bias capturing as discussed in section 4.2. Specifically, as the bias prevalence of the training distribution becomes higher, better bias capture can be achieved even without our method, thus making our improvement on the performance less significant, but still quite effective. This conclusion is supported by our experimental results shown in Figure 5.
>
> - **Variance between algorithms**: this is likely due to the fact that BEL is also a method targeted to enhance the bias capture procedure of the debiasing framework. As we can see that BEL is much more robust to the change in the bias magnitude and prevalence from Table 1. In other words, certain overlap between the goals of BEL and our method resulted in a smaller improvement of our method on the BEL baselines.
>
>
>
> As for quantifying the feature complexity of vision and NLP data, it is an extremely challenging and open research problem. To the best of our knowledge, a qualitative assessment is the standard in this domain. [1,2,3]
>
>
>
> [1] On the foundations of shortcut learning, Hermann et al, ICLR 2024 Oral
>
> [2] Entropy is not enough for test-time adaptation: From the perspective of disentangled factors, Lee et al., ICLR 2024 Spotlight
>
> [3] Learning from failure: Debiasing classifier from biased classifier, Nam et al, NeurIPS 2020
>
> ## W4: Theory-practice gap (Binary vs. Multi-class)
>
> We apologize for the confusion. **Our theoretical propositions are in fact already based on a multi-class setting**. As defined on line 243, we state: "the target attribute $y^t \sim \{a_1^t, ..., a_n^t\}$ and a spurious attribute $y^s \sim \{a_1^s, ..., a_m^s\}$". The analysis focuses on a pair of target feature $a_i^t$ and spurious feature $a_j^s$ each time, but is not restricted to binary attributes .
>
> ## W5: Coverage of related work (PGD)
>
> Thank you for pointing out this omission. We have included a discussion of PGD in the related work section of our revised paper. Specifically, Per-sample Gradient-based Debiasing (PGD) is a two-stage method that identifies bias-conflicting (BC) samples by their high gradient norms from an auxiliary biased model, then resamples the data to focus the final debiased model on these "hard" samples. Despite different technique routes, it's assumed by PGD that the biases can be well captured by the biased models, which serves as a foundation for subsequent debiasing based on gradients. However, in this work, we show that such assumptions might be challenged under real-world scenarios.
>
>
> Again, we’d like to thank the reviewer for their time and effort devoted to the reviewing process. **All revisions of the paper are marked in blue for your review.**

---

### Author Response · Authors · 2025-11-27
**A gentle reminder**

Dear Reviewers,

We thank you again for the time and effort you dedicated to reviewing our paper.

We wanted to gently follow up to ensure you have had a chance to read our rebuttal. We have addressed all of your questions and concerns point-by-point and are standing by to answer any final questions you might have to ensure a comprehensive evaluation of our work.

As the discussion period is coming to a close soon, we are eager to hear your thoughts. If you have any remaining questions or require further clarification, please let us know so we can respond before the deadline.

Best regards,
The Authors

---

### Author Response · Authors · 2025-12-02
**Summary of Discussion - Part 1**

To the Area Chair,

We sincerely thank you for managing the review process and the reviewers for their constructive feedback. We are encouraged that the reviewers unanimously recognized the novelty of our problem framing and the effectiveness of our proposed solution. During the rebuttal period, we have actively engaged with all reviewers, clarifying that significant empirical evidence was present in our original Appendix and conducting extensive additional experiments (including new tabular datasets) to fully address remaining concerns.

Below is a summary of the strengths identified by the reviewers and how we have decisively addressed their concerns.

## Summary of Strengths

The reviewers consistently highlighted the paper’s strong motivation, novel theoretical perspective, and the simplicity and effectiveness of the proposed method (DiD).

- **Novel Problem Framing & Analysis:** Reviewers praised our challenge to existing assumptions.
  - **Reviewer RjaM:** Commended the "**Clear problem framing**" contrasting benchmarks with reality and the "**Rigorous theory**" justifying the sparsity of real-world biases.
  - **Reviewer SKNU:** Highlighted the "**Novel problem framing and fine-grained bias analysis**" noting that distinguishing bias magnitude and prevalence provides a "**strong foundation for future benchmark design.**"
  - **Reviewer k3gy:** Stated the work tackles an "**Important problem**" with a "**Novel analytical perspective**" that is "**misaligned with reality**" in current benchmarks.
- **Methodological Simplicity & Effectiveness:** The proposed solution (DiD) was well-received.
  - **Reviewer SKNU:** Praised the "**Simplicity and generality of the proposed DiD method**," noting it is "**conceptually simple**" and "**empirically improves performance.**"
  - **Reviewer o4es:** Described DiD as a "**simple yet effective... approach that generalizes well across multiple datasets and modalities.**"
- **Evaluation & Benchmarking:**
  - **Reviewer o4es:** Appreciated the "**comprehensive empirical and theoretical analysis.**"
  - **Reviewer k3gy:** Noted the "**Thorough experiments**" across multiple baselines and the value of **RDBench**, stating that releasing it "**will benefit the community.**"

---

> ### Author Response · Authors · 2025-12-02
> **Summary of Discussion - Part 2**
>
> ## Summary of Concerns and Resolutions
> We have addressed all reviewer concerns, resulting in a significantly strengthened manuscript. Here we summarize the major concerns from the reviewers along with our resolutions.
> ### **Concern 1: Validation on Real-World Datasets (Reviewers RjaM, SKNU, o4es)**
> Reviewers expressed concern regarding the representation of real-world data, with Reviewer SKNU specifically noting the absence of COMPAS and MS COCO experiments despite their mention in the introduction. We address the concern as follows:
> - **Clarification of Existing Results:** We clarified that extensive experiments on large-scale real-world datasets—specifically **CelebA** (Vision), **MultiNLI**, and **CivilComments-WILDS** (NLP)—were indeed included in the original submission. However, due to strict page limits, these results were deferred to the **Appendix**, which likely caused them to be overlooked during the initial review. We have now brought these results to the forefront to demonstrate that the method was already validated on diverse real-world benchmarks.
> - **Additional Tabular Experiments (Addressing Reviewer SKNU):** To directly address Reviewer SKNU’s final comment regarding the specific lack of tabular data, we conducted new experiments during the rebuttal on **three real-world tabular datasets**: COMPAS, Adult, and German. Our method (DID) consistently improved performance over strong baselines (LfF, DisEnt, BEL, BED) across all three.
>
> In conclusion, with the clarification of existing Appendix results and the addition of new tabular experiments, our evaluation now **covers 12 datasets, 6 of which are real-world datasets, across 3 modalities** (Vision, NLP, and Tabular), rigorously verifying real-world applicability. Thus, we believe the comprehensiveness of experiments are in fact **a notable strength of the paper rather than a weakness, significantly exceeding that of recent works**.
>
> ||Venue|MNIST-based|CIFAR-based|WaterBirds|BFFHQ|BAR|CelebA|NICO|MultiNLI|CCW|COMPAS|Adult|German|
> |-|-|-|-|-|-|-|-|-|-|-|-|-|-|
> |DeNetDM|NIPS 24|√|√||√|√|√|||||||
> |DPR|NIPS 24|√|||√|√|√|||√||||
> |BCSI|NIPS 24|√|√||√|||√|√|√||||
> |**Ours**||**√**|**√**|**√**|**√**|**√**|**√**|**√**|**√**|**√**|**√**|**√**|**√**|
> ### **Concern 2: Clarification of "Unsupervised" Nature and Thresholds (Reviewers RjaM, SKNU)**
> Reviewers questioned the use of thresholds (extreme threshold of 0) and whether the method relied on prior knowledge of bias attributes. We address the concern as follows:
> - We clarified that the **thresholds are strictly analytical tools** used to characterize datasets in our paper, not hyperparameters for the algorithm. And more importantly, our core finding, demonstrated in Figure 2(b), is that **real-world datasets consistently show much lower bias prevalence than benchmarks across a range of *meaningful* thresholds** (e.g., 0.1), which *remains true* regardless of the behavior at the extreme 0 value.
> - We emphasized that **DiD is fully unsupervised and bias-agnostic**. It does not require bias labels or prevalence estimation during training. The experiments demonstrate effectiveness across various distribution types (LMLP, HMLP, HMHP) without prior knowledge.
> ### **Concern 3: Theoretical Clarifications (Reviewer RjaM)**
> Reviewer RjaM asked for better theoretical exposition regarding multi-class settings.
>
> We revised Section 3 and the Appendix proofs to clarify that **our theory is build upon the multi-class setting** (not just binary).
> ### **Concern 4: Related Work Discussion (Reviewers RjaM, o4es)**
> Reviewers asked for comparisons to specific works like PGD and DiffuBias.
>
> We added discussions and comparisons to **PGD** (Reviewer RjaM) and **DiffuBias** (Reviewer SKNU) in the related work section, positioning our method against these approaches.
> ### **Concern 5: Computational Overhead Analysis (Reviewers SKNU, k3gy)**
> Reviewers SKNU and k3gy requested an analysis of the computational complexity and overhead introduced by our method. We address the concern as follows:
> - We clarified that DiD is implemented as a simple **data preprocessing plugin** (a data transformation applied only to the batch fed to the biased auxiliary model).
> - We confirmed that the method adds **negligible computational overhead** in terms of both training time and memory, as the overall complexity remains dominated by the base debiasing backbone.
> ## Conclusion
>
> We believe we have sufficiently addressed the concerns raised by the reviewers. By clarifying that key real-world results were originally located in the Appendix and supplementing them with new tabular experiments (COMPAS, Adult, German), we have closed the gap identified by Reviewer SKNU. The paper now presents a robust, thoroughly validated framework and solution that challenges the development of debiasing methods.
>
> We respectfully request that you consider these improvements and the strong consensus on the paper's novelty in your final decision.
>
> Best regards,
>
> The Authors

---

### Meta-Review · Area_Chair_Ut9H · 2026-01-10

**Summary:**

This paper received good comments and a number of serious concerns. Scoring was 4, 4, 6, 6.

First of all, a raised remark regards the justifications of the work, which are not enough elaborated and convincing. Other issues regard a theory-practice gap, lack on implementation details, doubts about the unsupervised nature of the task, and many issues of the experimental phase in general.

The latter involves flaws in the design (e.g., it's unclear how the different data distributions LMLP and HMLP are composed), limited real-world evidence in validation, performance variation not sufficiently justified, lack of failure case analysis, missing comparisons, experiments in general limited in diversity and scalability.

Insufficient discussion about interpretability and heavy use of heuristics (e.g., in hyper-parameters' setting) are also critical points. Computational cost is also not discussed and SotA is also considered not up-to-date.

**Reviewer Concerns:**

The rebuttal addresses all comments and seems elaborated. Since main issues regard the experimental analysis, authors also add new results on real datasets (mainly on the supplementary material), but it is unclear if improvements reach the state of the art, it does not seem so.

Other comparative results are provided adopting C-CIFAR10 e C-MNIST as dataset, but they also do not appear to reach state of the art despite these data are considered quite simple. Results for COCO e COMPAS datasets are also added. In summary, if state-of-the art performance in absolute terms is reached remains unclear.

Overall, the rebuttal seems quite extensive, but not fully convincing.

All comments are addressed, new results are given, which were among the main raised concerns, but the quality of such results is difficult to assess. The gap between theory and practice does not seem to be filled. Paper revision is quoted, but it's negligible except for the new results, meaning that the requests for clarification do not seem to be well addressed.

**Reviewer Scores:**

A short discussion occurs just with Reviewer SKNU, which is not finalized, but new results for COCO and COMPAS were included, as requested.

Rev. o4es wrote a short review with a positive score, but it seems a bit mismatched with the reported comments.

I do not expect that reviewers who scored 4 (below threshold) will increase their ratings.

---

### Decision · Program_Chairs · 2026-01-26

Reject